

# The Interactions between Soil-Biosphere-Atmosphere (ISBA) land surface model Multi-Energy Balance (MEB) option in SURFEX - Part 1: Model description

Aaron Boone[1], Patrick Samuelsson[2], Stefan Gollvik[2], Adrien Napoly[1],
Lionel Jarlan[3], Eric Brun[1], and Bertrand Decharme[1]

[1]CNRM UMR 3589, Météo-France/CNRS, Toulouse, France
[2]Swedish Meteorological and Hydrological Institute, Norrköping, Sweden
[3]CESBIO UMR 5126, Toulouse, France

*Correspondence to:* Aaron Boone (aaron.a.boone@gmail.com)

**Abstract.**

Land surface models (LSMs) are pushing towards improved realism owing to an increasing number of observations at the local scale, constantly improving satellite data-sets and the associated methodologies to best exploit such data, improved computing resources, and in response to the user

community. As a part of the trend in LSM development, there have been ongoing efforts to improve the representation of the land surface processes in the Interactions between the Surface Biosphere Atmosphere (ISBA) LSM within the EXternalized SURFace (SURFEX) model platform.

The Force-Restore approach in ISBA has been replaced in recent years by improved realism with respect to for example, multi-layer explicit physically-based options for sub-surface heat transfer,

soil hydrological processes, and the composite snowpack. The representation of vegetation processes in SURFEX has also become much more sophisticated in recent years, including photosynthesis and respiration and biochemical processes. It become clear that the conceptual limits of the composite soil-vegetation scheme within ISBA have been reached and there is a need to explicitly separate the canopy vegetation from the soil surface. In response to this issue, a collaboration began in 2008

between the High-Resolution Limited Area Model (HIRLAM) consortium and Météo-France with the intention to develop an explicit representation of the vegetation in ISBA under the SURFEX platform. A new parameterization has been developed called the ISBA Multi-Energy Budget (MEB) in order to address these issues. ISBA-MEB consists in a fully-implicit numerical coupling between a multi-layer physically-based snowpack model, a variable-layer soil scheme, an explicit litter layer,

a bulk vegetation scheme, and the atmosphere. It also includes a feature which permits a coupling transition of the snowpack from the canopy air to the free atmosphere. It shares many of the routines and physics parameterizations with the standard version of ISBA. This paper is the first of two parts:



in part one, the ISBA-MEB model equations, numerical schemes and theoretical background are presented. In part two which is a separate companion paper, a local scale evaluation of the new

scheme is presented along with a detailed description of the new forest litter scheme.

## 1    Introduction

Land Surface Models (LSMs) are based upon fundamental mathematical laws and physics applied within a theoretical framework. Certain processes are modeled explicitly while others use more conceptual approaches. They are designed to work across a large range of spatial scales, so that

unresolved scale-dependent processes represented as a function of some grid-average state variable using empirical or statistical relationships. LSMs were originally implemented in numerical weather prediction (NWP) and global climate models (GCMs) in order to provide interactive lower boundary conditions for the atmospheric radiation and turbulence parameterization schemes over continental land surfaces. In the past two decades, LSMs have evolved considerably to include more biogeo-

chemical and biogeophysical processes in order to meet the growing demands of both the research and the user communities (Pitman, 2003; van den Hurk et al., 2011). A growing number of state-of-the-art LSMs which are used in coupled atmospheric models for operational numerical weather prediction (Ek et al., 2003; Boussetta et al., 2013), climate modeling (Oleson et al., 2010; Zhang et al., 2015), or both (Best et al., 2011; Masson et al., 2013), represent most or all of the following pro-

cesses: photosynthesis and the associated Carbon fluxes, multi-layer soil water and heat transfer, vegetation phenology and dynamics (biomass evolution, net primary production), sub-grid lateral water transfer, river routing, atmosphere-lake exchanges, snow pack dynamics, and near surface urban meteorology. Some LSMs also include processes describing the Nitrogen cycle (Castillo et al., 2012), groundwater exchanges (Vergnes et al., 2014), aerosol surface emissions (Cakmur et al., 2004), iso-

topes (Braud et al., 2005), and the representation of human impacts on the hydrlgical cycle in terms of irrigation (de Rosnay et al., 2003) and ground water extraction (Pokhrel et al., 2015), to name a few.

As a part of the trend in LSM development, there have been ongoing efforts to improve the representation of the land surface processes in the Interactions between the Surface Biosphere Atmosphere

(ISBA) LSM within the EXternalized SURFace (SURFEX: Masson et al.,2013) model platform. The original two-layer ISBA Force-Restore model (Noilhan and Planton, 1989) consists in a single bulk soil layer (generally having a thickness on the order of 50 cm to several m) coupled to a superficially thin surface composite soil-vegetation-snow layer. Thus, the model simulates so-called fast processes which occur at sub-diurnal timescales which are pertinent to short term numerical weather

prediction, and it provides a longer term water storage reservoir which provides a source for transpiration, a time filter for water reaching a hydro-graphic network, and a certain degree of soil moisture memory in the ground amenable to longer term forecasts and climate modeling. Additional modifi-



cations were made to this scheme over the last decade to include soil freezing (Boone et al., 2000; Giard and Bazile, 2000), improved hydrolgical processes (Mahfouf and Noilhan, 1996; Boone et al., 1999; Decharme and Douville, 2006). This scheme was based on the pioneering work of Deardorff (1977) and it has proven its value for coupled land-atmosphere research and applications since its inception. For example, it is currently used for research within the Mesoscale Non-Hydrostatic research model (Meso-NH) (Lafore et al., 1998), operational high resolution short term numerical weather prediction at Météo-France within the limited area model AROME (Seity et al., 2011) and by HIRLAM countries within the ALADIN-HIRLAM system as the HARMONIE-AROME model configuration (Bengtsson et al., 2016), and climate research within the global climate model (GCM) Action de Researche Petite Echelle Grande Echelle (ARPEGE-climat: Voldoire et al.,2013) and by HIRLAM countries within the ALADIN-HIRLAM system as HARMONIE-AROME and HARMONIE-ALARO Climate configurations (Lind et al., 2016).

## 1.1 Rationale for improved vegetation processes

Currently, many LSMs are pushing towards improved realism owing to an increasing number of observations at the local scale, constantly improving satellite data-sets and the associated methodologies to best exploit such data, improved computing resources, and in response to the user community via climate services (and seasonal forecasts, drought indexes, etc...). In the SURFEX context the Force-Restore approach has been replaced in recent years by improved realism with respect to for example, multi-layer explicit physically-based options for sub-surface heat transfer (Boone et al., 2000; Decharme et al., 2016), soil hydrological processes (Boone et al., 2000; Decharme et al., 2011, 2016), and the composite snowpack (Boone and Etchevers, 2001; Decharme et al., 2016), These new schemes have recently been implemented in the operational distributed hydrometeorological hindcast system SAFRAN-ISBA-MODCOU (SIM) (Habets et al., 2008), Meso-NH, and ARPEGE-climat and ALADIN-HIRLAM HARMONIE-AROME and HARMONIE-ALARO Climate configurations. The representation of vegetation processes in SURFEX has also become much more sophisticated in recent years, including photosynthesis and respiration (Calvet et al., 1998), Carbon allocation to biomass pools (Calvet and Soussana, 2001; Gibelin et al., 2006), and soil carbon cycling (Joetzjer et al., 2015). However, for a number of reasons it has also become clear that we have reached the conceptual limits of using of a composite soil-vegetation scheme within ISBA and there is a need to explicitly separate the canopy vegetation from the soil surface:

– Vegetation canopy, ground and snow surface temperatures can have very different amplitudes and phases in terms of the diurnal cycle. Accounting for this distinction the incorporation of remote-sensing data into such models, such as using satellite-based thermal infrared data (e.g., Anderson et al., 1997)



- it has become evident that the only way to simulate the snowpack beneath forests in a robust and a physically consistent manner (i.e. lessening the dependence of forest snow cover on highly empirical and poorly constrained snow fractional cover parameterizations, among other things) and including certain key processes (such as canopy interception and unloading of snow) is to include a forest canopy above or buried by the ground-based snowpack (e.g., Rutter et al., 2009)

- for accurately modeling canopy radiative transfer, within or below canopy turbulent fluxes and soil heat fluxes

- to make a more consistent photosynthesis and Carbon allocation model (including explicit Carbon stores for the vegetation, litter and soil in a consistent manner)

- to allow the explicit treatment of a ground litter layer, which has a significant impact on ground heat fluxes and soil temperatures (and freezing), and by extension, the turbulent heat fluxes.

In response to this issue, a collaboration began in 2008 between the High-Resolution Limited Area Model (HIRLAM) consortium and Météo-France with the intention to develop an explicit representation of the vegetation in ISBA under the SURFEX platform. A new parameterization has been developed called the ISBA Multi-Energy Budget (MEB) in order to account for all of the above issues.

MEB is based on the classic two-source model for snow-free conditions which considers explicit energy budgets (for computing fluxes) for the soil and the vegetation, and it has been extended to a three-source model in order to include an explicit representation of snowpack processes and their interactions with the ground and the vegetation. The vegetation canopy is represented using the so-called big-leaf method which lumps the entire vegetation canopy into a single effective leaf for computing energy budgets and the associated fluxes of heat, moisture and momentum. One of the first examples of a two-source model designed for atmospheric model studies is Deardorff (1978), and further refinements to the vegetation canopy processes were added in the years that followed leading to fairly sophisticated schemes which are similar to those used today (e.g., Sellers et al., 1986). The two-source big-leaf approach has been used extensively within coupled regional and global scale land-atmosphere models (Xue et al., 1991; Sellers et al., 1996; Dickinson et al., 1998; Lawrence et al., 2011; Samuelsson et al., 2011). In addition, more recently multi-layer vegetation schemes have also been developed for application in GCMs (Bonan et al., 2014; Ryder et al., 2016).

ISBA-MEB has been developed taking the same strategy which has been used historically for ISBA: inclusion of the key first order processes while maintaining a system which has minimal input data requirements and computational cost while being consistent with other aspects of ISBA (with the ultimate goal of being used in coupled operational numerical weather forecast and climate models, and spatially distributed monitoring and hydrological modeling systems). In 2008, one of the HIRLAM partners, the Swedish Meteorological and Hydrological Institute (SMHI), had



already developed and applied an explicit representation of the vegetation in the Rossby Centre
Regional Climate Model (RCA3) used at SMHI (Samuelsson et al., 2006, 2011). This representa-
tion was introduced into the operational NWP HIRLAMv7.3 system which became operational in
2010. In parallel, the dynamic vegetation model LJP-GUESS was coupled to RCA3 as RCA-GUESS
Smith et al. (2011) making it possible to simulate complex biogeophysical feedback mechanisms in
climate scenarios. Since then RCA-GUESS has been applied over Europe Wramneby et al. (2010),
Africa Wu et al. (2016) and the Arctic Zhang et al. (2014). The basic principles developed by SMHI
has been the foundation when the explicit representation of the vegetation has been introduced in
ISBA and SURFEX, but now in a more general and consistent way. Implementation of canopy turbu-
lence scheme, longwave radiation transmission function and snow interception formulations in MEB
largely follows the implementation done in RCA3 (Samuelsson et al., 2006, 2011). In addition, we
have taken this opportunity to incorporate several new features into ISBA-MEB compared to the
original SMHI scheme:

- a snow fraction which can gradually bury the vegetation vertically thereby transitioning the
  turbulence coupling from the canopy air space directly to the atmosphere (using a fully implicit
  numerical scheme)

- the use of the detailed solar radiation transfer scheme which is a multi-layer model that con-
  siders two spectral bands, direct and diffuse flux components and the concept of sunlit and
  shaded leaves and was developed to improve the modeling of photosynthesis within ISBA
  (Carrer et al., 2013)

- a more detailed treatment of canopy snow interception and unloading processes and a coupling
  with the ISBA physically-based multi-layer snow scheme,

- a reformulation of the turbulent exchange coefficients within the canopy air space for stable
  conditions, such as over a snowpack

- a fully implicit Jacobean matrix for the longwave fluxes from multiple surfaces (snow, below-
  canopy snow-free ground surface, vegetation canopy)

- all of the energy budgets are numerically implicitly coupled with each other and with the at-
  mosphere using the coupling method adapted from Best et al. (2004) which was first proposed
  by Polcher et al. (1998).

- an explicit forest litter layer model (which also acts as the below-canopy surface energy budget
  when litter covers the soil)

This paper is the first of two parts: in part one, the ISBA-MEB model equations, numerical
schemes and theoretical background are presented. In part two, a local scale evaluation of the new





scheme is presented along with a detailed description of the new forest litter scheme (Napoly et al., 2016). An overview of the model is given in the next section, followed by conclusions.

## 2    Model Description

SURFEX uses the tile approach for the surface, and separate physics modules are used to compute
surface-atmosphere exchange for oceans or seas, lakes, urbanized areas and the natural land sur-
face (Masson et al., 2013). The ISBA LSM is used for the latter tile, and the land surface is further
split into upwards of 12 or 19 so-called patches (refer to Table 1) which represent the various land
cover and plant functional types. Currently, forests make up 8 patches for the 19-class option, and
three for the 12-class option. ISBA-MEB (referred to hereafter simply as MEB) option can be ac-
tivated for any number of the forest patches. By default, MEB is coupled to the multi-layer soil
(DIF) (Boone et al., 2000; Decharme et al., 2011), and snow (ES) schemes (Boone and Etchevers,
2001; Decharme et al., 2016). These schemes have been recently updated (Decharme et al., 2016)
to include improved physics and increased layering (12 layers for both by default). MEB can also
be coupled to the simple 3-layer soil Force-Restore (3-L) option (Boone et al., 1999) in order to be
compatible with certain applications which have historically used 3-L, but by default, it is coupled
with DIF since the objective is to move towards a less conceptual LSM.

A schematic diagram for a maximum illustrating the various resistance pathways for the turbulent
fluxes for the three fully (implicitly) coupled surface energy budgets is shown in Fig. 1. The wa-
ter budget prognostic variables are also indicated. There are six aerodynamic resistance, $R_a$ (s$^{-1}$),
pathways which are indicated in red and defined as being between; i) the non-snow buried vege-
tation canopy and the canopy air, $R_{avg-c}$, ii) the non-snow buried ground surface (soil or litter)
and the canopy air, $R_{ag-c}$, iii) the snow surface and the canopy air, $R_{an-c}$, iv) the ground-based
snow-covered part of the canopy and the canopy air, $R_{avn-c}$, v) the canopy air with the overlying at-
mosphere, $R_{ac-a}$), and vi) the ground-based snow surface (directly) with the overlying atmosphere,
$R_{an-a}$. Previous papers describing ISBA (Noilhan and Planton, 1989; Mahfouf and Noilhan, 1991)
expressed heat fluxes using a dimensionless heat and mass exchange coefficient, $C_H$: however for
the new MEB option, it is more convenient to express the different fluxes using resistances (s m$^{-1}$)
which are related to the exchange coefficient as $R_a = 1/(V_a\,C_H)$.

The surface energy budgets are formulated in terms of prognostic equations for the temperature
evolution of the bulk vegetation canopy, $T_v$, the snow-free ground surface (soil or litter), $T_g$, and the
ground-based snowpack, $T_n$ (K: indicated in magenta). The prognostic hydrological variables are in
blue and represent the liquid soil water content, $W_g$, liquid water equivalent ice content, $W_{gf}$, snow
water equivalent (SWE), $W_n$, vegetation canopy intercepted liquid water, $W_r$, and intercepted snow,
$W_{rn}$ (kg m$^{-2}$). The diagnosed variables which are determined implicitly during the simultaneous
solution of the energy budgets are colored in dark red; the surface specific humidity at saturation for





each of the three energy budgets, $q$ (kg kg$^{-1}$), and the canopy air specific humidity, $q_c$, temperature, $T_c$ and wind speed, $V_c$ (m s$^{-1}$). The surface snow cover fraction area is represented by $p_{ng}$, while the fraction of the canopy buried by the ground-based snowpack is defined as $p_{\alpha n}$ The snowpack has $N_n$ layers, while the number of soil layers is defined as $N_g$ where $k$ is the vertical index (increasing from

1 at the surface downward). The ground and snowpack uppermost layer temperatures correspond to those used for the surface energy budget (i.e. $k = 1$).

## 2.1   Snow Fractions

Snow is known to have a significant impact on heat conduction fluxes owing to it's relatively high insulating properties. In addition, it can significantly reduce turbulent transfer owing to reduced

surface roughness, and it has a relatively large surface albedo thereby impacting the surface net radiation budget. Thus, the parameterization of it's areal coverage turns out to be a critical aspect of LSM modeling of snowpack-atmosphere interactions and sub-surface soil and hydrological processes. The fractional ground coverage by the snowpack is defined as

$$p_{ng} = W_n/W_{n,crit} \qquad\qquad (0 \le p_{ng} \le 1) \qquad\qquad\qquad (1)$$

where currently the default value is $W_{n,crit} = 1$ (kg m$^{-2}$). Note that this is considerably lower than the previous value of 10 kg m$^{-2}$ used in ISBA (Douville et al., 1995), but this value has been shown to improve the ground soil temperatures using an explicit snow scheme within ISBA (Brun et al., 2013).

The fraction of the vegetation canopy which is buried by ground-based snow is defined as

$$p_{n\alpha} = (D_n - z_{hv,b})/(z_{hv} - z_{hv,b}) \qquad\qquad (0 \le p_{n\alpha} \le 1) \qquad\qquad\qquad (2)$$

where $D_n$ is the total ground-based snowpack depth (m), and $z_{hvb}$ represents the base of the vegetation canopy (m) (see Fig. 2) which is currently defined as

$$z_{hvb} = a_{hv}\,(z_{hv} - z_{hv,min}) \qquad\qquad (z_{hvb} \ge 0) \qquad\qquad\qquad (3)$$

where $a_{hv} = 0.2$ and the effective canopy base height is set to $z_{hv,min} = 2$ (m) for forests. The

foliage distribution should be reconsidered in further development since literature suggests, e.g. Massman (1982), that the foliage is not symmetrically distributed in the crown but skewed upward.



### 2.2 Energy Budget

The coupled energy budget equations for a three-source model can be expressed for a single bulk canopy, a ground-based snowpack and a underlying ground surface as

$$225 \qquad \mathcal{C}_v \frac{\partial T_v}{\partial t} = R_{nv} - H_v - LE_v + L_f \Phi_v \qquad (4)$$

$$\mathcal{C}_{g,1} \frac{\partial T_{g,1}}{\partial t} = (1 - p_{ng})(R_{ng} - H_g - LE_g) + p_{ng}(G_{gn} + \tau_{n,N_n} SW_{nn}) - G_{g,1} + L_f \Phi_{g,1} \quad (5)$$

$$p_{ng} \mathcal{C}_{n,1} \frac{\partial T_{n,1}}{\partial t} = (R_{nn} - H_n - LE_n - \tau_{n,1} SW_{nn} + \xi_{n,1} - G_{n,1} + L_f \Phi_{n,1}) p_{ng} \qquad (6)$$

where $T_{g,1}$ is the uppermost ground (surface soil or litter layer) temperature, $T_{n,1}$ is the surface
snow temperature, and $T_v$ is the bulk-canopy temperature (K). Note that the subscript 1 indicates the uppermost layer or the base of the layer (for fluxes) for the soil and snowpack. $L_f$ represents the latent heat of fusion, and $LE$ represents the various latent heat flux terms. The ground-based snow fraction is defined as $p_{ng}$. Note that the terms of Eq. 6 are multiplied by $p_{ng}$ to make them patch-relative (or grid-box relative in the case of single-patch mode) since the snow can potentially
cover only part of the patch. Within the snow module itself, the notion of $p_{ng}$ is not used (the computations are snow-relative). The formulation for $p_{ng}$ is described in Section 2.1. The phase change terms (freezing less melting: expressed in kg m$^{-2}$ s$^{-1}$) terms for the snow water equivalent intercepted by the vegetation canopy, the uppermost ground layer, and the uppermost snowpack layer are represented by $\Phi_v$, $\Phi_{g,1}$ and $\Phi_{n,1}$, respectively. The computation of $\Phi_{g,1}$ uses the Gibbs
free-energy method (Decharme et al., 2016), $\Phi_{n,1}$ is based on available liquid for freezing or cold content for freezing (Boone and Etchevers, 2001), and $\Phi_v$ is described herein (see Eq. 83). Note that all of the phase change terms are computed as adjustments to the surface temperatures (after the fluxes have been computed), therefore only the energy storage terms are modified directly by phase changes for each model time step. The last term on the right-hand-side (RHS) of Eq. 6, $\xi_{n,1}$,
represents the effective heating or cooling of a snowpack layer caused by exchanges in enthalpy between the surface and sub-surface model layers when the vertical grid is reset (the snow model grid layer thicknesses vary in time).

The surface ground, snow, and vegetation effective heat capacities, $\mathcal{C}_{g,1}$, $\mathcal{C}_v$ and $\mathcal{C}_{n,1}$ (J m$^{-2}$ K$^{-1}$) are defined, respectively, as

$$250 \qquad \mathcal{C}_{g,1} = \Delta z_{g,1}\, c_{g,1} \qquad (7)$$

$$\mathcal{C}_v = C_{vb} + C_i W_{r,n} + C_w W_r \qquad (8)$$

$$\mathcal{C}_{n,1} = D_{n,1}\, c_{n,1} \qquad (9)$$

where $C_i$ and $C_w$ are the specific heat capacities for solid ($2.106 \times 10^3$ J kg$^{-1}$ K$^{-1}$) and liquid
water ($4.218 \times 10^3$ J kg$^{-1}$ K$^{-1}$), respectively. The uppermost ground layer thickness is $\Delta z_{g,1}$ (m), and the corresponding heat capacity of this layer is defined as $c_{g1}$ (J m$^{-3}$ K$^{-1}$). The uppermost soil layer ranges between 0.01 and 0.03 m for most applications, so that the interactions between



surface fluxes and fast temperature changes in the surface soil layer can be represented. There are
two options for modeling the thermal properties of the uppermost ground layer. First, they can be

defined using the default ISBA configuration for a soil layer with parameters based on soil texture
properties which can also incorporate the thermal effects of soil organics (Decharme et al., 2016).
The second option, which is the default when using MEB, is to model the uppermost ground layer as
forest litter. The ground surface in forest regions is generally covered by a litter layer consisting of
dead leaves and or needles, branches, fruit, and other organic material. Some LSMs have introduced

parameterizations for litter (Enrique et al., 1999; Ogée and Brunet, 2002; Wilson et al., 2012), but
the approach can be very different from one to another depending on their complexity. The main goal
of this parameterization within MEB is to account for the generally-accepted first-order energetic
and hydrological effects of litter; this layer is generally accepted to have a strong insulating effect
owing to its particular thermal properties (leading to a relatively low thermal diffusivity), it causes

a significant reduction of ground evaporation (capillary rise into this layer is negligible), and it
constitutes an interception reservoir for liquid water which can also lose water by evaporation. See
Napoly et al. (2016) for a detailed description of this scheme and it's impact on the surface energy
budget.

The canopy is characterized by low heat capacity which means that its temperature responds fast to

changes in fluxes. Thus, to realistically simulate diurnal variations in 2-meter temperature this effect
must be accounted for. Sellers et al. (1986) defined the value as being the heat capacity of 0.2 kg
$m^{-2}$ of water per unit leaf area index ($LAI$: $m^2$ $m^{-2}$). This results in values on the order of $1 \times 10^4$
J $m^{-2}$ $K^{-1}$ for forest canopies in general. For local scale simulations, $C_{vb}$ can be defined based on
observational data. In spatially distributed simulations (or when observational data is insufficient),

$C_{vb} = 0.2/C_V$ where the vegetation thermal inertia, $C_V$ is defined as a function of vegetation class
by the SURFEX default physiographic database ECOCLIMAP (Faroux et al., 2013). Note that $C_V$
has been determined for the composite soil-vegetation scheme, so the factor 0.2 is used to reduce this
value to be more representative of vegetation and on the order of the value discussed by Sellers et al.
(1986). Numerical tests have shown that using this value, the canopy heat storage is on the order of

10 W $m^{-2}$ at mid-day for a typical mid-latitude summer day for a forest. The minimum vegetation
heat capacity value is limited at $1 \times 10^4$ (J $m^{-2}$ $K^{-1}$) in order to model, in a rather simple fashion,
the thermal inertia of stems, branches, trunks, etc. The contributions from intercepted snow and rain
are incorporated, where $W_{r,n}$ and $W_r$ (kg $m^{-2}$) represent the equivalent liquid water content of
intercepted canopy snow and liquid water, respectively.

The uppermost snow layer thickness is $D_{n,1}$ (m), and the corresponding heat capacity is repre-
sented by $c_{n,1}$ (Boone and Etchevers, 2001). Note that $D_{n,1}$ is limited to values no larger than several
centimeters in order to model a reasonable thermal inertia (i.e. in order to represent the diurnal cycle)
in a fashion analogous to the soil. For more details, see Decharme et al. (2016).




The numerical solution of the surface energy budget, sub-surface soil and snow temperatures, and
the implicit numerical coupling with the atmosphere is described in Appendix I.

### 2.3 Turbulent fluxes

In this section, the turbulent heat and water vapor fluxes in Eq.s 4-6 are described.

#### 2.3.1 Sensible heat fluxes

The MEB sensible heat fluxes are defined as

$$H_v = \rho_a \frac{(\mathcal{T}_v - \mathcal{T}_c)}{R_{a\,v-c}} \tag{10}$$

$$H_g = \rho_a \frac{(\mathcal{T}_g - \mathcal{T}_c)}{R_{a\,g-c}} \tag{11}$$

$$H_n = \rho_a \left[ (1 - p_{n\alpha}) \frac{(\mathcal{T}_n - \mathcal{T}_c)}{R_{a\,n-c}} + p_{n\alpha} \frac{(\mathcal{T}_n - \mathcal{T}_a)}{R_{a\,n-a}} \right] \tag{12}$$

$$H_c = \rho_a \frac{(\mathcal{T}_c - \mathcal{T}_a)}{R_{a\,c-a}} \tag{13}$$

$$H = \rho_a \left[ (1 - p_{n\alpha} p_{ng}) \frac{(\mathcal{T}_c - \mathcal{T}_a)}{R_{a\,c-a}} + p_{n\alpha} p_{ng} \frac{(\mathcal{T}_n - \mathcal{T}_a)}{R_{a\,n-a}} \right] \tag{14}$$

where $\rho_a$ represents the lowest atmospheric layer average air density (kg m$^{-3}$). The fluxes between
the canopy air space and the vegetation, $H_v$, the snow-free ground, $H_g$, and the ground-based snow-
pack, $H_n$, appear in the surface energy budget equations (Eq.s4-6). The sensible heat flux from the
ground-based snowpack (Eq. 12) is partitioned by the fraction of the vegetation which is buried
by the ground-based snowpack, $p_{n\alpha}$, between an exchange between the canopy air space, and the
overlying atmosphere (Eq. 2). The heat flux between the overlaying atmosphere and the canopy air
space is represented by $H_c$, and it is equivalent to the sum of the fluxes between the different energy
budgets and the canopy air space. The total flux exchange between the overlying atmosphere and the
surface (as seen by the atmosphere) is defined by $H$. It is comprised of two components: the heat ex-
change between the overlying atmosphere and the canopy air space and the part of the ground-based
snowpack which is burying the vegetation. This method has been developed to model the covering
of low vegetation canopies by a ground-based snowpack. Finally, the final fluxes for the given patch
are aggregated using $p_{ng}$ and $p_{n\alpha}$: the full expressions are given in Appendix C1.

The thermodynamic variable ($\mathcal{T}$: J kg$^{-1}$) is linearly related to temperature as

$$\mathcal{T}_x = \mathcal{B}_x + \mathcal{A}_x T_x \tag{15}$$

where $x$ corresponds to one of the three surface temperatures, canopy air temperature, $T_c$, or the
overlying atmospheric temperature, $T_a$. The definitions of $\mathcal{A}_x$ and $\mathcal{B}_x$ depend on the atmospheric
variable in the turbulent diffusion scheme and are usually defined to cast $\mathcal{T}$ in the form of dry static
energy, or potential temperature and are determined by the atmospheric model in coupled mode (see
Appendix A).





The total canopy aerodynamic resistance is comprised of snow-buried, $R_{avn-c}$, and non-snow buried, $R_{avg-c}$, resistances from

$$R_{av-c} = \left[ \frac{(1-p_{n\alpha})\,p_{ng}}{R_{avn-c}} + \frac{(1-p_{ng})}{R_{avg-c}} \right]^{-1} \tag{16}$$

The separation of the resistances is done to mainly account for differences in the roughness length
between the buried and non-covered parts of the vegetation canopy, so the primary effect of snow cover is to increase the resistance relative to a snow-free surface assuming the same temperature gradient owing to a lower surface roughness, thus $R_{avn-c} \geq R_{avg-c}$. The formulation also provides a continuous transition to the case of vanishing canopy turbulent fluxes as the canopy becomes entirely buried (as $p_{n\alpha} \to 1$). In this case, the energy budget equations collapse into a simple coupling
between the snow surface and the overlying atmosphere, and the ground energy budget is simply consists in heat conduction between the ground surface and the snowpack base. The formulations of the resistances between the different surfaces and the canopy airspace and the overlying atmosphere are described in detail in Sect. 2.6. The canopy air temperature, which is needed by different physics routines, is diagnosed by combining Eq. s10-14 and solving for $\mathcal{T}_c$ and using Eq. 15 to determine $T_c$
(see Appendix A for details).

### 2.3.2   Water vapor fluxes

The MEB water vapor fluxes are expressed as

$$E_v = \rho_a\, h_{sv}\, \frac{(q_{satv} - q_c)}{R_{av-c}} \tag{17}$$

$$E_g = \rho_a\, \frac{(q_g - q_c)}{R_{ag-c}} \tag{18}$$

$$E_n = \rho_a\, h_{sn} \left[ (1-p_{n\alpha})\, \frac{(q_{satin} - q_c)}{R_{an-c}} + p_{n\alpha}\, \frac{(q_{satin} - q_a)}{R_{an-a}} \right] \tag{19}$$

$$E_c = \rho_a\, \frac{(q_c - q_a)}{R_{ac-a}} \tag{20}$$

$$E = \rho_a \left[ (1-p_{n\alpha}p_{ng})\, \frac{(q_c - q_a)}{R_{ac-a}} + p_{n\alpha}p_{ng}\, h_{sn}\, \frac{(q_{satin} - q_a)}{R_{an-a}} \right] \tag{21}$$

where, in an analogous fashion to the sensible heat flux, the vapor flux between the canopy air space
and the vegetation canopy, $E_v$, the snow-free ground, $E_g$, and the ground-based snowpack, $E_n$, correspond to the fluxes in the surface energy budgets (Eq.s 4-6). The vapor flux between the canopy air and the overlying atmosphere is represented by $E_c$, and the total vapor flux exchanged with the overlying atmosphere is defined as $E$. The specific humidity (kg kg$^{-1}$) of the overlying atmosphere is represented by $q_a$, while $q_{sat}$ and $q_{sati}$ represent the specific humidity at saturation over liquid
water and ice, respectively. For the surface specific humidities at saturation, the convention $q_{satx} = q_{sat}(T_x)$ is used. The canopy air specific humidity, $q_c$, is diagnosed assuming that $E_c$ is balanced by the vapor fluxes between the canopy air and each of the three surfaces considered (the methodology





for diagnosing the canopy air thermal properties is described in Appendix I, Section I3). The effective ground specific humidity is defined as

$$q_g = h_{sg}\, q_{sat\,g} + (1 + h_a)\, q_c \qquad\qquad (22)$$

where the so-called humidity factors are defined as

$$h_{sg} = \delta_g\, h_{ug}\, (1 - p_{gf}) \left(\frac{L_v}{L}\right) + \delta_{gf}\, h_{ugf}\, p_{gf} \left(\frac{L_s}{L}\right) \qquad\qquad (23)$$

$$h_a = \delta_g\, (1 - p_{gf}) \left(\frac{L_v}{L}\right) + \delta_{gf}\, p_{gf} \left(\frac{L_s}{L}\right) \qquad\qquad (24)$$

The latent heats of fusion and vaporization are defined as $L_s$ and $L_v$ (J kg$^{-1}$), respectively. The fraction of the surface layer which is frozen, $p_{gf}$, is simply defined as the ratio of the liquid water equivalent ice content to the total water content. The average latent heat, $L$, is essentially a normalization factor which ranges between $L_s$ and $L_v$ as a function of snow cover and surface soil ice (see Appendix B). The soil coefficient $\delta_g$ in Eq.s 23-24 is defined as

$$\delta_g = \left(\frac{R_{a\,g-c}}{R_{a\,g-c} + R_g}\right) \delta_{gcor} \qquad\qquad (25)$$

where the soil resistance, $R_g$, is defined by Eq. 67. Note that the composite version of ISBA did not include an explicit soil resistance term, so this also represents a new addition to the model This term was found to further improve results for baresoil evaporation within MEB, and it's inclusion is consistent with other similar multi-source models (e.g., Xue et al., 1991). See Sect. 2.6 for further

details. The delta function, $\delta_{gcor}$, is a numerical correction term which is required owing to the linearization of $q_{sat\,g}$ and is unity unless both $h_{ug}\, q_{sat\,g} < q_c$ and $q_{sat\,g} > q_c$, in which case it is set to zero. The surface ground humidity factor is defined using the standard ISBA formulation from Noilhan and Planton (1989) as

$$h_{ug} = \frac{1}{2} \left[ 1 - \cos\left(\frac{w_{g,1}}{w_{fc,1}^*}\pi\right) \right] \qquad (0 \le h_{ug} \le 1) \qquad\qquad (26)$$

In the case of condensation ($q_{sat\,g} < q_a$), $h_{ug} = 1$ (see Mahfouf and Noilhan, 1991, for details). The effective field capacity, $w_{fc,1}^*$ is computed relative to the liquid water content of the uppermost soil layer (it is adjusted in the presence of soil ice compared to the default field capacity). The analogous form holds for the humidity factor over the frozen part of the surface soil layer, $h_{ugf}$, with $w_{g,1}$ and $w_{fc,1}^*$ replaced by $w_{gf,1}$ and $w_{fcf,1}^*$ (m$^3$ m$^{-3}$) in Eq. 26, respectively (Boone et al., 2000). Note that

it would be more accurate to use $q_{sati}$ in place of $q_{sat}$ for the sublimation of the canopy-intercepted snow and the soil ice in Eq.s 17-18, respectively, but this complicates the linearization and this has been neglected for now. The snow factor is defined as $h_{sn} = L_s/L$. This factor can be modified so that $E_n$ includes both sublimation and evaporation (Boone and Etchevers, 2001), but the impact of including a liquid water flux has been found to be negligible thus for simplicity, only sublimation is

accounted for currently.





The leading coefficient for the canopy evapotranspiration is defined as

$$h_{sv} = (1 - p_{nv}) \, h_{svg} \, (L_v/L) + p_{nv} \, h_{svn} \, (L_s/L) \tag{27}$$

where $p_{nv}$ is defined by Eq. 79). When part of the vegetation canopy is buried (i.e. $p_{n\alpha} > 0$), a different roughness and $LAI$ are felt by the canopy air space so that a new resistance is computed over the $p_{n\alpha}$ covered part of the canopy as is done for sensible heat flux. This is accounted for by defining

$$h_{svg} = p_{ng} (1 - p_{n\alpha}) \left( \frac{R_{a\,v-c}}{R_{a\,vn-c}} \right) h_{vn} + (1 - p_{ng}) \left( \frac{R_{a\,v-c}}{R_{a\,vg-c}} \right) h_{vg} \tag{28a}$$

$$h_{svn} = p_{ng} (1 - p_{n\alpha}) \left( \frac{R_{a\,v-c}}{R_{a\,vn-c}} \right) + (1 - p_{ng}) \left( \frac{R_{a\,v-c}}{R_{a\,vg-c}} \right) \tag{28b}$$

The so-called Halstead coefficients in Eq. 28a are defined as

$$h_{vg} = \left( \frac{R_{a\,vg-c}}{R_{a\,vg-c} + R_s} \right) (1 - \delta) + \delta \tag{29a}$$

$$h_{vn} = \left( \frac{R_{a\,vn-c}}{R_{a\,vn-c} + R_{sn}} \right) (1 - \delta) + \delta \;, \tag{29b}$$

The stomatal resistance, $R_s$, can be computed using either the so-called Jarvis method (Jarvis, 1976) described by Noilhan and Planton (1989) or a more physically based method which includes a representation of photosynthesis (Calvet et al., 1998). The stomatal resistance for the partially snow-buried portion defined as

$$R_{sn} = R_s / \left[ 1 - \min \left( p_{n\alpha}, \, 1 - R_s/R_{s,max} \right) \right] \qquad (R_{sn} \leq R_{s,max}) \tag{30}$$

so that the effect of coverage by the snowpack is to increase the canopy resistance. Note that when the canopy is not partially or fully buried by ground based snowpack ($p_{n\alpha} = 0$) and does not contain any intercepted snow ($p_{nv} = 0$), the leading coefficient for the canopy evapotranspiration simplifies to the Halstead coefficient from the composite version of ISBA (Mahfouf and Noilhan, 1991)

$$h_{sv} = \left( \frac{R_{a\,vg-c}}{R_{a\,vg-c} + R_s} \right) (1 - \delta) + \delta \qquad (p_{n\alpha} = 0 \text{ and } p_{nv} = 0) \tag{31}$$

The fraction of the vegetation covered by water is $\delta$ and is described in Sect. 2.8.2.

The evapotranspiration from the vegetation canopy, $E_v$, is comprised of three components:

$$E_v = E_{tr} + E_r + E_{rn} \tag{32}$$

where the transpiration, evaporation from the canopy liquid water interception store and sublimation from the canopy snow interception store are represented by $E_{tr}$, $E_r$, and $E_{rn}$, respectively. The expressions for these fluxes are given in Appendix C.



### 2.4 Radiative fluxes

The $R_n$ terms in Eq.s 4-6 represent the surface net radiation terms (longwave and shortwave components):

$$R_{nx} = SW_{net,x} + LW_{net,x} \tag{33}$$

where $x = n, g$ or $v$. The total net radiation of the surface is

$$R_n = R_{nn} + R_{ng} + R_{nv} = SW\downarrow - SW\uparrow + LW\downarrow - LW\uparrow \tag{34}$$

where the total down-welling solar (shortwave) and atmospheric (longwave) radiative fluxes (W m$^{-2}$) at the top of the canopy or snow surface (in the case snow is burying the vegetation) are represented by $SW\downarrow$ and $LW\downarrow$, respectively. The total upwelling (towards the atmosphere) shortwave and longwave radiative fluxes, $SW\uparrow$ and $LW\uparrow$, respectively, are simply defined as the downward components less the total surface net radiative fluxes (summed over the three surfaces). The effective total surface albedo and surface radiative temperature (and emissivity) can then be diagnosed (see the Sect. 2.4.2) for coupling with the host atmospheric model. The $\tau_n$ is defined as the solar radiation transmission at the base of a snowpack layer, so that $\tau_{n,1} SW_{nn}$ term in Eq. 6 corresponds the amount of shortwave radiation which is not absorbed in the uppermost snowpack layer. For sufficiently thin snowpack, solar energy penetrating the snow to the underlying ground surface is expressed as $\tau_{n,N_n} SW_{nn}$, where $N_n$ represents the number of modeled snowpack layers (for a deep snowpack, this term becomes negligible).

#### 2.4.1 Shortwave Radiative Fluxes

The total land surface shortwave energy budget can be shown to satisfy

$$SW\downarrow = SW_{netg} + SW_{netv} + SW_{netn} + SW\uparrow \tag{35}$$

where $SW_{netg}$, $SW_{netv}$, $SW_{netn}$ represent the net shortwave terms for the ground, vegetation canopy and the ground-based snowpack. The effective surface albedo (which may be required by the atmospheric radiation scheme or for comparison with satellite-based data etc.) is diagnosed as

$$\overline{\alpha}_s = SW\uparrow / SW\downarrow \tag{36}$$

The multi-level transmission computations for direct and diffuse radiation are from Carrer et al. (2013). The distinction between the visible (VIS) and near-infrared (NIR) radiation components is important in terms of interactions with the vegetation canopy. Here, we take into account two spectral bands for the soil and the vegetation, where visible wavelengths range from approximately 0.3 to 0.7 m$^{-6}$, and near-infrared wavelengths range from approximately 0.7 to 1.4 m$^{-6}$. The spectral values for the soil and the vegetation are provided by ECOCLIMAP (Faroux et al., 2013) as a function of vegetation type and climate.





The effective all-wavelength ground (below-canopy) albedo is defined as

$$\overline{\alpha}_{gn} = p_{ng}\,\alpha_n + (1 - p_{ng})\,\alpha_g \tag{37}$$

where $\alpha_g$ represents the ground albedo. The ground-based snow albedo, $\alpha_n$, is prognostic and de-
455 pends on the snow grain size. It currently includes up to three spectral bands (Decharme et al., 2016),
however, when coupled to MEB, only the two aforementioned spectral bands are currently consid-
ered for consistency with the vegetation and soil.

The effective canopy albedo, $\overline{\alpha}_v$, represents the combined canopy vegetation, $\alpha_v$, and intercepted
snow albedos. Currently, however, we assume that $\overline{\alpha}_v = \alpha_v$ which is based on recommendations by
460 Pomeroy and Dion (1996). They showed that multiple reflections and scattering of light from patches
of intercepted snow together with a high probability of reflected light reaching the underside of an
overlying branch implied that trees actually act like light traps. Thus, they concluded that intercepted
snow had no significant influence on the short-wave albedo or the net radiative exchange of Boreal
conifer canopies.

In addition to baseline albedo values required by the radiative transfer model for each spectral
band, the model requires the direct and diffusive downwelling solar components. The diffuse fraction
can be provided by observations (offline mode) or a host atmospheric model. For the case when no
diffuse information is provided to the surface model, the diffuse fraction is computed using the
method proposed by Erbs et al. (1982).

**2.4.2 Longwave Radiative Fluxes**

The longwave radiation scheme is based on a representation of the vegetation canopy as a plane-
parallel surface. The model considers one reflection with three reflecting surfaces (ground, ground-
based snowpack and the vegetation canopy: a schematic is shown in Appendix E). The total land
surface longwave energy budget can be shown to satisfy

$$LW \downarrow = LW_{net\,g} + LW_{net\,v} + LW_{net\,n} + LW \uparrow \tag{38}$$

where $LW_{net\,g}$, $LW_{net\,v}$, $LW_{net\,n}$ represent the net longwave terms for the ground, vegetation
canopy and the ground-based snowpack. The effective surface radiative temperature (which may
be required by the atmospheric radiation scheme or for comparison with satellite-based data etc.) is
diagnosed as

$$T_{rad} = \left[\frac{LW \uparrow - LW \downarrow (1 - \overline{\epsilon}_s)}{\overline{\epsilon}_s\,\sigma}\right]^{1/4} \tag{39}$$

where $\sigma$ is the Stefan-Boltzmann constant, and $\overline{\epsilon}_s$ represents the effective surface emissivity. In
Eq. 39, there are two knowns ($LW$ fluxes) and two unknowns ($T_{rad}$ and $\overline{\epsilon}_s$). Here we opt to pre-
define $\overline{\epsilon}_s$ in a manner which is consistent with the various surface contributions as

$$\overline{\epsilon}_s = p_{ng}\,\overline{\epsilon}_{sn} + (1 - p_{ng})\,\overline{\epsilon}_{sg} \tag{40}$$





The canopy-absorption weighted effective snow and ground emissivities are defined, respecitvely, as

$$\overline{\epsilon}_{sn} = \overline{\sigma}_{n\,LW}\,\epsilon_v + (1 - \overline{\sigma}_{n\,LW})\,\epsilon_n \tag{41}$$

$$\overline{\epsilon}_{sg} = \overline{\sigma}_{g\,LW}\,\epsilon_v + (1 - \overline{\sigma}_{g\,LW})\,\epsilon_g \tag{42}$$

where $\epsilon_v$, $\epsilon_g$ and $\epsilon_n$ represent the emissivities of the vegetation, snow-free ground and the ground-based snowpack, respectively. The ground and vegetation emissivities are given by ECOCLIMAP for spatially distributed simulations, or they can be prescribed for local scale studies. The snow emissivity is currently defined as $\epsilon_n = 0.99$. The effect of longwave absorption through the non-snow buried part of the vegetation canopy is included as

$$\overline{\sigma}_{n\,LW} = [1 - p_{ng} - p_{n\alpha}\,(1 - p_{ng})]\,\sigma_{LW} + [p_{ng} + p_{n\alpha}\,(1 - p_{ng})]\,\sigma_{f\,LW} \tag{43}$$

$$\overline{\sigma}_{g\,LW} = [1 - p_{ng}\,(1 - p_{n\alpha})]\,\sigma_{LW} + p_{ng}\,(1 - p_{n\alpha})\,\sigma_{f\,LW} \tag{44}$$

where the canopy absorbtion is defined as

$$\sigma_{LW} = 1 - \exp\,(-\tau_{LW}\,LAI) = 1 - \chi_v \tag{45}$$

and $\tau_{LW}$ represents a longwave radiation transmission factor which can be species (or land classification) dependent, and $\chi_v$ is defined as a vegetation view factor. The absorption over the under-story snow-covered fraction of the grid box is modeled quite simply from Eq. 45 as

$$\sigma_{f\,LW} = 1 - \exp\,[-\tau_{LW}\,LAI\,(1 - p_{n\alpha})] = 1 - \exp\,[-\tau_{LW}\,LAI_n] \tag{46}$$

so that transmission is unity (no absorption or reflection by the canopy: $\overline{\sigma}_{LW} = \sigma_{f\,LW} = 0$) when $p_{n\alpha} = 1$ (i.e. when the canopy has been buried by snow). $LAI_n$ is used to represent the $LAI$ which has been reduced owing to burial by the snowpack. From Eq.s 40-44, it can be seen that when there is no snowpack (i.e. $p_{ng} = 0$ and $p_{n\alpha} = 0$), then the effective surface emissivity is simply an absorption-weighted soil-vegetation value defined as $\overline{\epsilon}_s = \sigma_{LW}\,\epsilon_v + (1 - \sigma_{LW})\,\epsilon_g$. See Appendix E for the derivation of the net longwave radiation terms in Eq. 38.

## 2.5 Heat Conduction fluxes

The sub-surface snow and ground heat conduction fluxes are modeled using Fourier's Law ($G = \lambda\,\partial T/\partial z$). The heat conduction fluxes in Eq.s 5-6 are written in discrete form as

$$G_{g,1} = \frac{2\,(T_{g,1} - T_{g,2})}{(\Delta z_{g,1}/\lambda_{g,1}) + (\Delta z_{g,2}/\lambda_{g,2})} = \Lambda_{g,1}\,(T_{g,1} - T_{g,2}) \tag{47}$$

$$G_{n,1} = \frac{2\,(T_{n,1} - T_{n,2})}{(D_{n,1}/\lambda_{n,1}) + (D_{n,2}/\lambda_{n,2})} = \Lambda_{n,1}\,(T_{n,1} - T_{n,2}) \tag{48}$$

$$G_{gn} = \frac{2\,(T_{n,N_n} - T_{g,1})}{(D_{n,N_n}/\lambda_{n,N_n}) + (\Delta z_{g,1}/\lambda_{g,1})} = \Lambda_{g,n}\,(T_{n,N_n} - T_{g,1}) \tag{49}$$

where $G_{gn}$ represents the snow-ground inter-facial heat flux which defines the snow scheme lower boundary condition. All of the internal heat conduction fluxes ($k = 2, N - 1$) use the same form





as in Eq. 48 for the snow (Boone and Etchevers, 2001) and Eq. 47 for the soil (Boone et al., 2000; Decharme et al., 2011). The heat capacities and thermal conductivities, $\lambda_g$, for the ground depend on the soil texture, organic content (Decharme et al., 2016) and potentially on the thermal properties of the forest litter in the uppermost layer (Napoly et al., 2016): all of the aforementioned properties depend on the water content. The snow thermal property parameterization is described in Decharme et al. (2016).

### 2.6 Aerodynamic Resistances

The resistances between the surface and the overlying atmosphere, $R_{an-a}$ and $R_{ac-a}$, are based on Louis (1979) modified by Mascart et al. (1995) to account for different roughness length values for heat and momentum as in ISBA: the full expressions are given in Noilhan and Mahfouf (1996).

#### 2.6.1 Aerodynamic Resistance between the bulk vegetation layer and the canopy air

The aerodynamic resistance between the vegetation canopy and the surrounding airspace can be defined as

$$R_{avg-c} = (g_{av} + g_{av}^*)^{-1} \tag{50}$$

The parameterization of the bulk canopy aerodynamic conductance, $g_{av}$, between the canopy and the canopy air is based on Choudhury and Monteith (1988). It is defined as

$$g_{av} = \frac{2\,LAI\,a_{av}}{\phi_v'} \left(\frac{u_{hv}}{lw}\right)^{1/2} [1 - \exp(-\phi_v'/2)]. \tag{51}$$

where $u_{hv}$ represents the wind speed at the top of the canopy (m s$^{-1}$), $LAI$ is the leaf area index (m$^2$ m$^{-2}$), and the remaining parameters are defined in Table 2. The conductance accounting for the free convection correction from Sellers et al. (1986) is expressed as

$$g_{av}^* = \left[\frac{LAI}{890}\left(\frac{T_v - T_c}{lw}\right)^{1/4}\right] \qquad (T_v \geq T_c) \tag{52}$$

Note that this correction is only used for unstable conditions. The effect of snow burying the vegetation impacts the aerodynamic resistance of the canopy is simply modeled by modifying the $LAI$ using

$$LAI_n = LAI\,(1 - p_{n\alpha}) \tag{53}$$

The $LAI_n$ is then used in Eq. 50 to compute $R_{avn-c}$, and this resistance is limited to 5000 s m$^{-1}$ as $LAI_n \to 0$.

#### 2.6.2 Aerodynamic Resistance between the ground and the canopy air

The resistance between the ground and the canopy air space is defined as

$$R_{ag-c} = R_{agn}/\psi_H \tag{54}$$





where $R_{agn}$ is the default resistance value for neutral conditions. The stability correction term, $\psi_H$, depends on the canopy structural parameters, wind speed and temperature gradient between the

550 surface and the canopy air. The aerodynamic resistance is also based on Choudhury and Monteith (1988). It is assumed that the eddy diffusivity, $K$ ($m^2\ s^{-1}$), in the vegetation layer follows an exponential profile:

$$K\left(z\right) = K\left(z_{hv}\right)\exp\left[\phi_v\left(1 - \frac{z}{z_{hv}}\right)\right] \tag{55}$$

where $z_{hv}$ represents the canopy height. Integrating the reciprocal of the diffusivity defined in Eq. 55

from $z_{0g}$ to $d + z_{0v}$ yields

$$R_{agn} = \frac{z_{hv}}{\phi_v\,K\left(z_{hv}\right)}\left\{\exp\left[\phi_v\left(1 - \frac{z_{0g}}{z_{hv}}\right)\right] - \exp\left[\phi_v\left(1 - \frac{d + z_{0v}}{z_{hv}}\right)\right]\right\} \tag{56}$$

The diffusivity at the canopy top is defined as

$$K\left(z_{hv}\right) = k\,u_{*hv}\left(z_{hv} - d\right) \tag{57}$$

The von Karman constant, $k$, has a value of 0.4. The displacement height is defined as (Choudhury and Monteith,

1988):

$$d = 1.1\,z_{hv}\ln\left[1 + \left(c_d\,LAI_f\right)^{1/4}\right] \tag{58}$$

where the leaf drag coefficient, $c_d$, is defined from Sellers et al. (1996):

$$c_d = 1.328\left[\frac{2}{R_e^{1/2}}\right] + 0.45\left[\frac{1}{\pi}(1 - \chi_L)\right]^{1.6} \tag{59}$$

$\chi_L$ represents the Ross-Goudriaan leaf angle distribution function, which has been estimated accord-

565 ing to Monteith (1975) (see Table 2), and $R_e$ is the Reynolds number defined as

$$R_e = \frac{u_l\,lw}{\upsilon}. \tag{60}$$

The friction velocity at the top of the vegetation canopy is defined as

$$u_{*hv} = \frac{k\,u_{hv}}{\ln\left[\left(z_{hv} - d\right)/z_{0v}\right]} \tag{61}$$

where the wind speed at the top of the canopy is

570 $$u_{hv} = f_{hv}\,V_a \tag{62}$$

and $V_a$ represents the wind speed at the reference height, $z_a$, above the canopy. The canopy height is defined based on vegetation class and climate within ECOCLIMAP as a primary parameter. It can also be defined using an external dataset, such as from a satellite-derived product (as a function of space and time). The vegetation roughness length for momentum is then computed as a secondary



parameter as a function of the vegetation canopy height. The factor $f_{hv}$ ($\leq 1$) is a stability dependent adjustment factor (see Appendix D).

The dimensionless height scaling factor is defined as

$$\phi_z = \frac{(z_{hv} - d)}{z_r} \qquad (\phi_z \leq 1) \tag{63}$$

The reference height is defined as $z_r = z_a - d$ for simulations where the reference height is suffi-
ciently above the top of the vegetation canopy. This is usually the case for local scale studies using observation data. When MEB is coupled to an atmospheric model, however, the lowest model level can be below the canopy height, so for coupled model simulations $z_r = \max(z_a, z_{hv} - d + z_{min})$ where $z_{min} = 2$ (m).

Finally, the stability correction factor from Eq. 54 is defined as

$$\psi_H = (1 - a_{hv} R_i)^{1/2} \qquad (R_i \leq 0) \tag{64a}$$

$$= \frac{1}{1 + b R_i (1 + c R_i)^{1/2}} \left[ 1 + \left( \frac{R_i}{R_{i,crit}} \right) (f_{z0} - 1) \right] \qquad (R_i > 0 \text{ and } R_i \leq R_{i,crit}) \tag{64b}$$

$$= \frac{f_{z0}}{1 + b R_i (1 + c R_i)^{1/2}} \qquad (R_i > R_{i,crit}) \tag{64c}$$

where the Richardson number is defined as

$$R_i = \frac{-g z_{hv} (T_s - T_c)}{T_s u_{hv}{}^2} \tag{65}$$

Note that strictly speaking, the temperature factor in the denominator should be defined as $(T_s + T_c)/2$, but this has only a minor impact for our purposes. The so-called critical Richardson number, $R_{i,crit}$, is set to 0.2. This parameter has been defined assuming that some turbulent exchange is likely always present (even if intermittent), but it is recognized that eventually a more robust approach should be
developed for very stable surface layers (Galperin et al., 2007). The expression for unstable conditions (Eq. 64a) is from (Sellers et al., 1996) where the structural parameter is defined as $a_{hv} = 9$.

It is generally accepted that there is a need to improve the parameterization of the exchange coefficient for extremely stable conditions typically encountered over snow (Niu and Yang, 2004; Andreadis et al., 2009). Since the goal here is not to develop a new parameterization, we simply
modify the expression for stable conditions by using the standard function from ISBA. The standard ISBA stability correction for stable conditions is given by Eq. 64c where $b = 15$ and $c = 5$ (Noilhan and Mahfouf, 1996). The factor which takes into account differing roughness lengths for heat and momentum is defined as

$$f_{z0} = \frac{\ln (z_{hv}/z_{0g})}{\ln (z_{hv}/z_{0gh})} \tag{66}$$

where $z_{0gh}$ is the ground roughness length for scalars. The weighting function (i.e. ratio of $R_i$ to $R_{i,crit}$) in Eq. 64b is used in order to avoid a discontinuity at $R_i = 0$ (the roughness length factor effect vanishes at $R_i = 0$) in Eq. 64c. An example of Eq. 64c is shown in Fig. 3 using the $z_{0g}$ from





Table 2, and for $z_{0gh}/z_{0g}$ of 0.1 and 1.0. Finally, the resistance between the ground-based snowpack, $R_{a\,n-c}$, and the canopy air use the same expressions as for the aerodynamic resistance between the ground and the canopy air outlined herein, but with the surface properties of the snowpack (namely the roughness length and snow surface temperature).

### 2.6.3 Ground resistance

The soil resistance term is defined based on Sellers et al. (1992) as

$$R_g = \exp\left[a_{Rg} - b_{Rg}\left(\overline{w}_g/\overline{w}_{sat}\right)\right] \ . \tag{67}$$

The coefficients are $a_{Rg} = 8.206$ and $b_{Rg} = 4.255$, and the vertically averaged volumetric water content and saturated volumetric water content are given by $\overline{w}_g$ and $\overline{w}_{sat}$, respectively. The averaging is done from one to several upper layers. Indeed, the inclusion of an explicit ground surface energy budget makes it more conceptually straightforward to include a ground resistance compared to the original composite soil-vegetation surface. The ground resistance is often used as a surrogate for an additional resistance arising due to a forest litter layer, therefore the soil resistance is set to zero when the litter layer option is activated. Finally, the coefficients $a_{Rg}$ and $b_{Rg}$ were determined from a case study for a specific location, and could possibly be location dependent. But currently these values are used, in part, since the litter formulation is the default configuration for MEB for forests as it generally gives better surface fluxes (Napoly et al., 2016).

## 2.7 Water Budget

The governing equations for (water) mass for the bulk canopy, and surface snow and ground layers are written as

$$\frac{\partial W_r}{\partial t} = P_{rv} + \max\left(0, -E_{tr}\right) - E_r - D_{rv} - \Phi_v \tag{68}$$

$$\frac{\partial W_{rn}}{\partial t} = I_n - U_n - E_{rn} + \Phi_v \tag{69}$$

$$p_{ng}\frac{\partial W_{n,1}}{\partial t} = P_s - I_n + U_n + p_{ng}\left(P_r - P_{rv} + D_{rv} - F_{nl,1} - E_n + \Phi_{n,1} + \xi_{nl,1}\right) \tag{70}$$

$$\rho_w\Delta z_{g,1}\frac{\partial w_{g,1}}{\partial t} = \left(P_r - P_{rv} + D_{rv} - E_g\right)\left(1 - p_{ng}\right) + p_{ng}F_{nl,N_n} - R_0 - F_{g,1} - \Phi_{g,1} \tag{71}$$

$$\rho_w\Delta z_{g,1}\frac{\partial w_{gf,1}}{\partial t} = \Phi_{g,1} - E_{gf}\left(1 - p_{ng}\right) \tag{72}$$

where $W_r$ and $W_{rn}$ represent the vegetation canopy water stores: intercepted water, and the intercepted snow and frozen water (all in kg m$^{-2}$), respectively. $W_{n,1}$ represents the snow liquid water equivalent (SWE) for the uppermost snow layer of the multi-layer scheme. The soil liquid water and equivalent frozen water equivalent volumetric water content are defined as $w_g$ and $w_{gf}$, respectively (m$^3$ m$^{-3}$).

The interception reservoir, $W_r$, is modeled as single layer bucket, with losses represented by evaporation, $E_r$, and canopy drip, $D_{rv}$, of liquid water which exceeds a maximum holding capacity




(see Sect. 2.8.2 for details). Sources include condensation (negative $E_r$ and $E_{tr}$) and $P_{rv}$ which represents the intercepted precipitation. The positive part of $E_{tr}$ is extracted from the sub-surface soil layers as a function of soil moisture and a prescribed vertical root zone distribution (Decharme et al., 2016). This equation is the same as that used in ISBA, except for the addition of the phase change term, $\Phi_v$ (kg m$^{-2}$ s$^{-1}$). This term has been introduced owing to the introduction of an explicit canopy snow interception reservoir, $W_{rn}$: the canopy snow and liquid water reservoirs can exchange mass via this term which is modeled as melt less freezing. The remaining rainfall $(P_r - P_{rv})$ is partitioned between the snow-free and snow-covered ground surface, where $P_r$ represents the total grid-cell rainfall rate. The canopy snow interception is more complex, and represents certain baseline processes such as snow interception, $I_n$, and unloading, $U_n$: see Sect. 2.8.1 for details.

The soil water and snow liquid water vertical fluxes at the base of the surface ground and snow are represented, respectively, by $F_{g,1}$ using Darcy's Law and by $F_{nl,1}$ using a tipping-bucket scheme (kg m$^{-2}$ s$^{-1}$). The liquid water flux at the base of the snowpack, $F_{nl,N_n}$, is directed downward into the soil and consists in the liquid water in excess of the lowest model liquid water holding capacity. A description of the snow and soil schemes are given in (Boone and Etchevers, 2001) and (Decharme et al., 2011), respectively. $R_0$ is the so-called surface runoff. It accounts for sub-grid heterogeneity of precipitation, soil moisture and for when potential infiltration exceeds a maximum rate (Decharme and Douville, 2006). The soil liquid water equivalent ice content can have some losses owing to sublimation in the uppermost soil layer, $E_{gf}$, but it mainly evolves owing to phase changes from soil water freeze-thaw, $\Phi_g$. The remaining symbols in Eq.s 68-69 are defined and described in Sections 2.8.2 and 2.8.1.

### 2.8 Precipitation Interception

#### 2.8.1 Canopy snow interception

The intercepted snow mass budget is described by Eq. 69, while the energy budget is included as a part of the bulk canopy prognostic equation (Eq. 4). The positive mass contributions acting to increase intercepted snow on canopy are snowfall interception, $I_n$, water on canopy that freezes, $\Phi_v < 0$, and sublimation of water vapor to ice, $E_{rn} < 0$. Unloading, $U_n$, sublimation, $E_{rn} > 0$, and snow melt, $\Phi_v > 0$, are the sinks. All of the terms are in kg m$^{-2}$ s$^{-1}$. It is assumed that intercepted rain and snow can co-exist on the canopy. The intercepted snow is assumed to have the same temperature as the canopy, $T_v$, thus there is no advective heat exchange with the atmosphere which simplifies the equations. For simplicity, when intercepted water on the canopy freezes, it is assumed to become part of the intercepted snow.

The parameterization of interception efficiency is based upon Hedstrom and Pomeroy (1998). It determines how much snow is intercepted during the time step and is defined as

$$I_{n,v,0} = (W_{rn}^* - W_{rn})\left[1 - \exp\left(-k_{n,v} P_s \Delta t\right)\right] \tag{73}$$





where $W_{rn}{}^*$ is the maximum snow load allowed, $P_s$ the frozen precipitation rate and $k_{n,v}$ a proportionality factor. $k_{n,v}$ is a function of $W_{rn}{}^*$ and the maximum plan area of the snow-leaf contact area per unit area of ground, $C_{n,vp}$:

$$k_{n,v} = \frac{C_{n,vp}}{W_{rn}{}^*} \tag{74}$$

For a closed canopy, $C_{n,vp}$ would be equal to one, but for a partly open canopy it is described by the relationship:

$$C_{n,vp} = \frac{C_{n,vc}}{1 - C_c\, u_{hv}\, z_{hv} / (w_n\, J_n)} \tag{75}$$

where $C_{n,vc}$ is the canopy coverage per unit area of ground which can be expressed as $1 - \chi_v$ where $\chi_v$ is the sky-view factor (see Eq. 45), and $u_{hv}$ represents the mean horizontal wind speed at the

685 canopy top (Eq. 62) which corresponds to the height $z_{hv}$ (m). The characteristic vertical snow-flake velocity, $w_n$, is set to 0.8 m s$^{-1}$ (Isymov, 1971). $J_n$ is set to $10^3$ m which is assumed to represent the typical size of the mean forested down wind distance.

For calm conditions and completely vertically falling snowflakes, $C_{n,vp} = C_c$. For any existing wind, snow could be intercepted by the surrounding trees so that high wind speed increases intercep-

690 tion efficiency. Generally for open Boreal conifer canopies, $C_{n,vc} < C_{n,vp} < 1$. Under normal wind speed conditions (i.e. wind speeds larger than 1 m s$^{-1}$), $C_{n,vc}$ (and $C_{n,vp}$) values are usually close to unity.

The maximum allowed canopy snow load, $W_{rn}{}^*$, is a function of the maximum snow load per unit branch area, $S_{n,v}$ (kg m$^{-2}$), and the leaf area index:

$$W_{rn}{}^* = S_{n,v}\, LAI \tag{76}$$

where $S_{n,v}$ is defined as

$$S_{n,v} = \overline{S_{n,v}} \left( 0.27 + \frac{46}{\rho_{n,v}} \right) \tag{77}$$

and $\overline{S_{n,v}} = 6.3$ kg m$^{-2}$ as suggested by (Schmidt and Gluns, 1991) for spruce canopies. $\rho_{n,v}$ is the canopy snow density (kg m$^{-3}$) defined by the relationship:

$$\rho_{n,v} = 67.92 + 51.25 \exp\left[ (T_c - T_f) / 2.59 \right] \qquad (T_c \le T_{cmax}) \tag{78}$$

where $T_c$ is the canopy air temperature and $T_{cmax}$ is the temperature corresponding to the maximum snow density. Assuming a maximum snow density of 750 kg m$^{-3}$ and solving Eq. 78 for canopy temperature yields $T_{cmax} = 279.854$ K. This gives values of $S_{n,v}$ in the range 4-6 kg m$^{-2}$.

The water vapor flux between the intercepted canopy snow and the canopy air, $E_{rn}$ (Eq. C6),

includes the evaporative efficiency, $p_{nv}$. This effect was first described by (Nakai et al., 1999). In the ISBA-MEB parameterization, the formulation is slightly modified so that it approaches zero when there is no intercepted snow load:

$$p_{nv} = \frac{0.89\, S_{nv}{}^{0.3}}{1 + \exp\left[ -4.7 (S_{nv} - 0.45) \right]} \tag{79}$$



where $S_{nv}$ is the ratio of snow-covered area on the canopy to the total canopy area:

$$S_{nv} = \frac{W_{rn}}{W_{rn}^*} \qquad (0 \leq S_{nv} \leq 1) \tag{80}$$

A numerical test is performed to determine if the canopy snow becomes less than zero within one time-step due to sublimation. If this is true, then the required mass is removed from the underlying snowpack so that the intercepted snow becomes exactly zero during the time-step to ensure a high degree of mass conservation. Note that this adjustment is generally negligible.

The intercepted snow unloading, due to processes such as wind and branch bending, has to be estimated. Hedstrom and Pomeroy (1998) suggest an experimentally verified exponential decay in load over time, t, which is used in the parameterization;

$$U_{n,v} = I_{n,v,0} \exp(-U_{nL}t) = I_{n,v,0}\, c_{nL} \tag{81}$$

where $U_{nL}$ is an unloading rate coefficient (s$^{-1}$) and $c_{nL}$ the dimensionless unloading coefficient. Hedstrom and Pomeroy (1998) found that $c_{nL} = 0.678$ was a good approximation which, with a time step of 15 minutes, gives $U_{nL} = -4.498 \cdot 10^{-6}$ s$^{-1}$. A tuned value for the RCA-LSM from the Snow Model Intercomparison Project phase 2 (SnowMIP2) experiments (Rutter et al., 2009) is $U_{nL} = -3.4254 \times 10^{-6}$ s$^{-1}$ which has been adopted for MEB for now. All unloaded snow is assumed to fall to the ground where it is added to the snow storage on forest ground. Further, corrections to compensate for changes in the original LSM due to this new parameterization have been made for heat capacity, latent heat of vaporisation, evapotranspiration, snow storages and fluxes of latent heat.

Finally, canopy snow will partly melt if the temperature rises above the melting point and become intercepted water, where the intercepted (liquid and frozen) water phase change is simply proportional to the temperature:

$$\Phi_v = \frac{C_i W_{rn}}{L_f \tau_\Phi}\left(T_f - T_v\right) = \frac{C_i S_{nv} W_{rn}^*}{L_f \tau_\Phi}\left(T_f - T_v\right) \tag{82}$$

where $\Phi_v < 0$ signifies melting. $T_f$ represents the melting point temperature (273.15 K) and the characteristic phase change timescale is $\tau_\Phi$ (s). If it is assumed that the available heating during the time step for phase change is proportional to canopy biomass via the $LAI$ then Eq. 82 can be written (for both melt and refreezing) as

$$\Phi_v = S_{nv}\, k_{\Phi v}\left(T_f - T_v\right) \tag{83}$$

Note that if energy is available for melting, the phase change rate is limited by the amount of intercepted snow, and likewise freezing is limited by the amount of intercepted liquid water. The melting of intercepted snow within the canopy can be quite complex, thus currently the simple approach in Eq. 83 adopted herein. The phase change coefficient was tuned to a value of $k_{\Phi v} = 5.56 \times 10^{-6}$ kg m$^{-2}$ s$^{-1}$ K$^{-1}$ for the SNOWMIP2 experiments with the RCA-LSM. Currently, this value is the default for ISBA-MEB.




### 2.8.2 Canopy rain interception

The rain intercepted by the vegetation is available for potential evaporation which means that it has a strong influence on the fluxes of heat and consequently also on the surface temperature. The rate of change of intercepted water on vegetation canopy is described by Eq. 68. The rate that water is intercepted by the over-story (which is not buried by the ground-based snow) is defined as

$$P_{rv} = P_r \left(1 - \chi_v\right) \left(1 - p_{ng} p_{\alpha n}\right) \tag{84}$$

where $\chi_v$ is a view factor indicating how much of the precipitation that should fall directly to the ground (see Eq. 45). The over-story canopy drip rate, $D_{rv}$, is defined simply as the value of water in the reservoir which exceeds the maximum holding capacity

$$D_{rv} = \max\left(0, W_{rv} - W_{rv,max}\right)/\Delta t \tag{85}$$

where the maximum liquid water holding capacity is defined simply as

$$W_{rv,max} = c_{wrv} LAI \tag{86}$$

Generally speaking, $c_{wrv} = 0.2$ (Dickinson, 1984), although it can be modified slightly for certain vegetation cover. Note that Eq. 68 is first evaluated with $D_{rv} = 0$, and then the canopy drip is computed as a residual. Thus, the final water amount is corrected by removing the canopy drip or through-fall. This water can then become a liquid water source for the soil and the ground-based snowpack.

The fraction of the vegetation covered with water is defined as

$$\delta_v = \left(1 - \omega_{rv}\right) \left(\frac{W_r}{W_{r,max}}\right)^{2/3} + \frac{\omega_{rv} W_r}{\left(1 + a_{rv} LAI\right) W_{r,max} - a_{rv} W_r} \tag{87}$$

Delire et al. (1997) used the first term on the RHS of Eq. 87 for relatively low vegetation (Deardorff, 1978) and the second term for tall vegetation (Manzi and Planton, 1994). Currently in ISBA, a weighting function is used which introduces the vegetation height dependence using the roughness length as a proxy from

$$\omega_{rv} = 2 z_{0v} - 1 \qquad \left(0 \leq \omega_{rv} \leq 1\right) \tag{88}$$

where the current value for the dimensionless coefficient is $a_{rv} = 2$.

### 2.8.3 Halstead Coefficient

In the case of wet vegetation, the total plant evapotranspiration is partitioned between the evaporation of intercepted water, and transpiration via stomata by the so-called Halstead coefficient. In MEB, two such coefficients are used for the non-snow buried and buried parts of the vegetation canopy, $h_{vg}$ and $h_{vn}$ (Eq.s 29a and 29b, respectively). In MEB, the general form of the Halstead coefficient, as





defined in Noilhan and Planton (1989), is modified by introducing the factor $k_v$ to take into account the fact that saturated vegetation can transpire, i.e. when $\delta_v = 1$ (Bringfelt et al., 2001). Thus for MEB, we define $\delta = k_v \, \delta_v$. The intercepted water forms full spheres just touching the vegetation surface when $k_v = 0$ which allows full transpiration from the whole leaf surface. In contrast, $k_v = 1$ would represent a situation where a water film covers the vegetation completely and no transpiration is allowed. To adhere to the interception model as described above, where the intercepted water exists as droplets, we set the value of $k_v$ to 0.25. Note that in the case of condensation, i.e. $E < 0$, $h_v = 1$.

Without a limitation of $h_{vg}$ and $h_{vn}$, the evaporative demand could exceed the available intercepted water during a time step, especially for the canopy vegetation which experiences a relatively low aerodynamic resistance. To avoid such a situation, a maximum value of the Halstead coefficient is imposed by calculating a maximum value of the $\delta_v$. See Appendix F for details.

## 3 Conclusions

This paper presents the description of a new multi-energy budget (MEB) scheme for representing tall vegetation in the ISBA land surface model component of the SURFEX land-atmosphere coupling and driving platform. This effort is part of the ongoing effort within the international scientific community to continually improve the representation of land-surface processes for hydrological and meteorological research and applications.

MEB consists in a fully-implicit numerical coupling between a multi-layer physically-based snowpack model, a variable-layer soil scheme, an explicit litter layer, a bulk vegetation scheme, and the atmosphere. It also includes a feature which permits a coupling transition of the snowpack from the canopy air to the free atmosphere as a function of snow depth and canopy height using a fully implicit numerical scheme. MEB has been developed in order to meet the criteria associated with computational efficiency, high coding standards (especially in terms of modularity), conservation (of mass, energy and momentum), numerical stability for large (time step) scale applications, and state-of-the-art representation of the key land surface processes required for current hydrological and meteorological modeling research and operational applications at Météo-France and within the international community as a part of the HIRLAM consortium. This includes regional scale real-time hind-cast hydro-meteorological modeling, coupling within both research and operational non-hydrostatic models, regional climate models and a global climate model, not to mention being used for ongoing offline land-surface reanalysis projects and fundamental research applications.

The simple composite soil-vegetation surface energy budget approach of ISBA has proven it's ability to provide solid scientific results and realistic boundary conditions for hydrological and meteorological models since it's creation over two decades ago. However, owing to the ever-increasing demands of the user community, it was decided to improve the representation of the vegetation



processes as a priority. The key motivation of the MEB development was to move away from the composite scheme in order to address certain known issues (such as excessive bare-soil evaporation in forested areas, the neglect of canopy snow interception processes), to improve consistency in

terms of the representation of the Carbon cycle (by modeling explicit vegetation energy and Carbon exchanges), to add new key explicit processes (forest litter, the gradual covering of vegetation by ground-based snow cover), and to open the door to potential improvements in land data assimilation (by representing distinct surface temperatures for soil and vegetation). Finally, note that while some LSMs intended for GCMs now use multiple-vegetation layers, a single bulk vegetation layer is cur-

rently used in MEB since it has been considered as a reasonable first increase in complexity level from the composite soil-vegetation scheme. However, MEB has been designed such that the addition of more canopy layers could be added if deemed necessary in the future.

This is part one of two companion papers describing the model formulation of ISBA-MEB. Part two describes the model evaluation at the local scale for several contrasting well-instrumented sites

in France, and for over 42 sites encompassing a wide range of climate conditions for several different forest classes over multiple annual cycles (Napoly et al., 2016, this issue). This two-part set of papers will be followed by a series of papers in upcoming years which will present the evaluation and analysis of ISBA-MEB with a specific focus (coupling with snow processes, regional to global scale hydrology, and finally fully coupled runs in a climate model).

**4   Code Availability**

The MEB code is a part of the ISBA LSM and is available as open source via the surface modelling platform called SURFEX, which can be downloaded at http://www.cnrm-game-meteo.fr/surfex/. SURFEX is updated at a relatively low frequency (every 3 to 6 months) and the developments presented in this paper are available starting with SURFEX version 8.0. If more frequent updates are

830 needed, or if what is required is not in Open-SURFEX (DrHOOK, FA/LFI formats, GAUSSIAN grid), you are invited to follow the procedure to get a SVN account and to access real-time modifications of the code (see the instructions at the previous link).

*Acknowledgements.*  This work was initiated within the international HIRLAM consortium as part of the ongoing collective effort to improve the SURFEX platform for research and operational hydrological and meteoro-

835 logical applications. We wish to make a posthumous acknowledgement of the contribution to this work by Joël Noilhan, who was one of the original supporters of this project and helped initiate this endeavor: his scientific vision was essential at the early stages of this work. We wish to thank other contributors to this development in terms of discussions and evaluation, such as G. Boulet, E. Martin, J.-C. Calvet, P. Le Moigne, C. Canac, and G. Aouad. The technical support of S. Faroux of the SURFEX team is also greatly appreciated. We also

wish to thank E. Lebas for preparing the MEB schematic. Part of this work was supported by a grant from Météo-France.



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





### Appendix A: Thermodynamic coupling variable

If potential temperature is used as the thermodynamic variable in the coupled model diffusion scheme, then the thermodynamic variable, $\mathcal{T}$ (J kg$^{-1}$: see Eq.s 10-14) coefficients are defined as

$$\mathcal{B}_x = 0 \qquad\qquad (x = v, g, n, c, a) \qquad\qquad\qquad (A1)$$

$$\mathcal{A}_x = C_p / \Pi_s \qquad\qquad (x = v, g, n, c) \qquad\qquad\qquad (A2)$$

$$\mathcal{A}_a = C_p / \Pi_a \qquad\qquad\qquad\qquad\qquad\qquad (A3)$$

where $\Pi$ is the non-dimensional Exner function and $C_p$ is the heat capacity of dry air (J kg$^{-1}$ K$^{-1}$). If the atmospheric variable being diffused is dry static energy then

$$\mathcal{B}_x = 0 \qquad\qquad (x = v, g, n, c) \qquad\qquad\qquad (A4)$$

$$\mathcal{B}_a = g\, z_a \qquad\qquad\qquad\qquad\qquad\qquad (A5)$$

$$\mathcal{A}_x = C_p \qquad\qquad (x = v, g, n, c, a) \qquad\qquad\qquad (A6)$$

where $z_a$ is the height (m) of the simulated or observed overlying atmospheric temperature, $T_a$ and $g$ is the gravitational constant. The choice of the atmospheric thermodynamic variable is transparent to ISBA-MEB (it is made within the surface-atmosphere coupler). The default (in offline mode and in in-line mode with certain atmospheric models) is using Eq.s A1-A3. Note that the method can be extended to use the actual air heat capacity (including water vapor) if a linearization of the heat capacity is used.

### Appendix B: Latent heat normalization factor

The $L$ is a normalization factor ($L_v \leq L \leq L_s$) which could be determined in a number of ways. This coefficient ensures conservation of mass between the different surfaces and the atmosphere. One possible method is to diagnose it by inverting the equation for $LE_c$ (multiplying Eq. I15 by $L$ thereby eliminating it from the RHS of this equation, and then solving for $L$), but the resulting equation is difficult to apply since the terms can be either positive or negative, and division by a small number is possible. Here, a more smooth (in time) function is proposed which accounts for each of the surfaces weighted by it's respective fraction:

$$L = \frac{a_{Ls} L_s + a_{Lv} L_v}{a_{Ls} + a_{Lv}} \qquad\qquad\qquad\qquad (B1)$$

where

$$a_{Lv} = \left[ \sigma_f \left( 1 - p_{nv} \right) + \left( 1 - p_{ng} \right) \left( 1 - p_{gf} \right) \right] \left( 1 - p_{ng} p_{n\alpha} \right) \qquad (B2)$$

$$a_{Ls} = \left[ \sigma_f p_{nv} + \left( 1 - p_{ng} \right) p_{gf} + p_{ng} \right] \left( 1 - p_{ng} p_{n\alpha} \right) + p_{ng} p_{n\alpha} \qquad (B3)$$

In the limit as the snow totally buries the canopy vegetation, $L \to L_s$. In contrast, for snow and surface ice free conditions, $L = L_v$.



## Appendix C: Turbulent Flux expressions

The turbulent fluxes of heat and water vapor can be further decomposed into different components which are required for computing different diagnostics and coupling with the water budgets. They are presented herein.

### C1 Sensible heat flux

It is convenient to split $H_n$ into two components since one governs the coupling between the canopy
air space and the snow surface, while the other modulates the exchanges with the overlying atmosphere (as the canopy layer becomes buried). The ground-based snowpack heat flux, $H_n$ (Eq. 12), can be split into a part which modulates the heat exchange with the canopy air space, $H_{n-c}$ and the other part which controls the exchanges directly with the overlying atmosphere, $H_{n-a}$, defined as

$$H_{n-c} = \rho_a \frac{(\mathcal{T}_n - \mathcal{T}_c)}{R_{a\,n-c}} \tag{C1}$$

$$H_{n-a} = \rho_a \frac{(\mathcal{T}_n - \mathcal{T}_a)}{R_{a\,n-a}} \tag{C2}$$

$\mathcal{T}_c$ is diagnosed by imposing conservation of the heat fluxes between the surface and the canopy air (As described in Appendix I). Using the definition in Eq. C2, the total sensible heat flux exchange with the atmosphere (Eq. 14) can also be written in more compact form as

$$H = \rho_a \left[ (1 - p_{ng} p_{n\alpha}) H_c + p_{ng} p_{n\alpha} H_{n-a} \right] \tag{C3}$$

### C2 Water vapor flux

The various water vapor flux terms must be broken into different components for use within the different water balance equations for the vegetation, soil and snowpack. Using the definitions in Eq.s 27-29b, the components of the canopy evapotranspiration, $E_v$, can be expressed as 1

$$E_{tr} = \rho_a \left( \frac{L_v}{L} \right) (q_{sat\,v} - q_c) \left[ \frac{p_{ng}(1 - p_{n\alpha})}{R_{a\,vn-c} + R_{sn}} + \frac{1 - p_{ng}}{R_{a\,vg-c} + R_s} \right] (1 - p_{nv})(1 - \delta) \tag{C4}$$

$$E_r = \rho_a \left( \frac{L_v}{L} \right) (q_{sat\,v} - q_c) \left[ \frac{p_{ng}(1 - p_{n\alpha})}{R_{a\,vn-c}} + \frac{1 - p_{ng}}{R_{a\,vg-c}} \right] (1 - p_{nv}) \delta \tag{C5}$$

$$E_{rn} = \rho_a \left( \frac{L_s}{L} \right) (q_{sat\,v} - q_c) \left[ \frac{p_{ng}(1 - p_{n\alpha})}{R_{a\,vn-c}} + \frac{1 - p_{ng}}{R_{a\,vg-c}} \right] p_{nv} \tag{C6}$$

The complex resistances (bracketed terms in Eq.s C4-C6) arise owing to the inclusion of the effects
of burying the snow canopy by the ground based snowpack. If the ground-based snowpack is not sufficiently deep to bury any of the canopy ($p_{n\alpha} = 0$), then the bracketed term in Eq. C4 simplifies to $1/(R_{a\,vg-c} + R_s)$ (note that $R_{a\,vg-c} = R_{a\,v-c}$ when $p_{n\alpha} = 0$ from Eq. 16), and likewise the bracketed terms in Eq.s C5-C6 simplify to $1/R_{a\,vg-c}$. Finally, the partitioning between the vapor fluxes from intercepted snow and the snow-free canopy reservoir and transpiration is done using
$p_{nv}$, which represents the fraction of the snow interception reservoir which is filled (see Eq. 79).





Using the definitions of $q_g$ from Eq. 22 together with those for the humidity factors, $h_{sg}$ and $h_a$ (Eq.s 23 and 24, respectively) and the soil coefficient, $\delta_g$ (Eq. 25), the bare soil evaporation, $E_g$, components can be expressed as

$$E_{gl} = \rho_a \left( \frac{L_v}{L} \right) (h_{ug}\, q_{sat\,g} - q_c) \left( \frac{\delta_{gcor}}{R_{a\,g} + R_g} \right) (1 - p_{gf}) \tag{C7}$$

$$E_{gf} = \rho_a \left( \frac{L_s}{L} \right) (h_{ugf}\, q_{sat\,g} - q_c) \left( \frac{\delta_{gfcor}}{R_{a\,g} + R_{gf}} \right) p_{gf} \tag{C8}$$

where $E_g = E_{gl} + E_{gf}$. The delta function, $\delta_{gfcor}$, is a numerical correction term which is required owing to the linearization of $q_{sat\,g}$ and is unity unless both $h_{ugf}\, q_{sat\,g} < q_c$ and $q_{sat\,g} > q_c$, in which case it is set to zero. Note that the ground resistances, $R_g$ and $R_{gf}$, are set to zero if the forest litter option is active (the default for forests).

The ground-based snowpack sublimation, $E_n$ (Eq. 19), can be partitioned into a vapor exchange with the canopy air space, $E_{n-c}$ and the overlying atmosphere, $E_{n-a}$, as

$$E_{n-c} = \rho_a \left( \frac{L_s}{L} \right) \left( \frac{q_{sat\,n} - q_c}{R_{a\,n-c}} \right) \tag{C9}$$

$$E_{n-a} = \rho_a \left( \frac{L_s}{L} \right) \left( \frac{q_{sat\,n} - q_a}{R_{a\,n-a}} \right) \tag{C10}$$

The corresponding latent heat fluxes can be determined by simply multiplying Eq. C4-C8 by $L$. Finally, using the definition in Eq. C10, the total vapor exchange with the atmosphere (Eq. 21) can also be written in more compact form as

$$E = \rho_a \left[ (1 - p_{ng}\, p_{n\alpha})\, E_c + p_{ng}\, p_{n\alpha}\, E_{n-a} \right] \tag{C11}$$

## Appendix D: Canopy-top wind stability factor

The expressions for the stability factor $f_{hv}$ (Eq. 62) which is used to compute the wind at the top of the vegetation canopy, $u_{hv}$, are taken from Samuelsson et al. (2006, 2011). They are defined as

$$f_{hv} = (C_{v,N} + C_{v,S}) \sqrt{C_D} / k \qquad (R_i > 0) \tag{D1a}$$

$$= (C_{v,N} + C_{v,U}) \sqrt{C_D} / k \qquad (R_i \le 0) \tag{D1b}$$

where the Richardson number, $R_i$, is defined in Eq. 65. The coefficients are defined as

$$C_{v,N} = \ln\left\{ 1 + \phi_z \left[ \exp\left( \frac{k}{\sqrt{C_{DN}}} \right) - 1 \right] \right\} \tag{D2}$$

$$C_{v,S} = - \phi_z \left( \frac{k}{\sqrt{C_{DN}}} - \frac{k}{\sqrt{C_D}} \right) \tag{D3}$$

$$C_{v,U} = - \ln\left\{ 1 + \phi_z \left[ \exp\left( \frac{k}{\sqrt{C_{DN}}} - \frac{k}{\sqrt{C_D}} \right) - 1 \right] \right\} \tag{D4}$$

where the drag coefficient, $C_D$, and the drag coefficient for neutral conditions, $C_{DN}$, are computed between the canopy air space and the free atmosphere above using the standard ISBA surface layer transfer functions (Noilhan and Mahfouf, 1996).





**Appendix E: Longwave Radiative Flux expressions**

The complete expression for the vegetation canopy net longwave radiation with an infinite number of reflections can be expressed as a series expansion (e.g., Braud, 2000) as a function of the temperatures of the emitting surfaces ($T_v$, $T_{g,1}$, $T_{n,1}$), their respective emissivities ($\epsilon_v$, $\epsilon_g$ and $\epsilon_n$) and the canopy longwave absorption function, $\sigma_{LW}$ (Eq. 45). The MEB expressions are derived by explicitly expanding the series and assuming one reflection from each emitting source, which is a good approximation since emissivities are generally close to unity (fluxes from a single reflection are proportional to $1-\epsilon_x$ where $x$ represents $g$, $v$ or $n$, and $\epsilon$ is close to unity for most natural surfaces).

Snow is considered to be intercepted by the vegetation canopy and to accumulate on the ground below. The corresponding schematic of the radiative transfer is shown in Fig. 4. The canopy-intercepted snow is treated using a composite approach, so that the canopy temperature, $T_v$, represents the effective temperature of the canopy-intercepted snow composite. The canopy emissivity is therefore simply defined as

$$\overline{\epsilon}_v = (1 - p_{nv})\epsilon_v + p_{nv}\epsilon_n \tag{E1}$$

In order to facilitate the use of a distinct multi-layer snow process scheme, we split the fluxes between those interacting with the snowpack and the snow-free ground. The expressions for the snow-free surface are

$$A_g = LW \downarrow (1 - p_{ng}) \tag{E2a}$$

$$B_g = A_g\,\sigma_{LW}\,(1 - \overline{\epsilon}_v) \tag{E2b}$$

$$C_g = A_g\,(1 - \sigma_{LW}) \tag{E2c}$$

$$D_g = C_g\,(1 - \epsilon_g) \tag{E2d}$$

$$E_g = D_g\,(1 - \sigma'_{LW}) \tag{E2e}$$

$$F_g = \sigma'_{LW}\,\sigma\,\overline{\epsilon}_v\,T_v^4\,(1 - p_{ng}) \tag{E2f}$$

$$G_g = F_g\,(1 - \epsilon_g) \tag{E2g}$$

$$H_g = G_g\,(1 - \sigma'_{LW}) \tag{E2h}$$

$$I_g = \sigma\,\epsilon_g\,T_g^4\,(1 - p_{ng}) \tag{E2i}$$

$$J_g = I_g\,\sigma'_{LW}\,(1 - \overline{\epsilon}_v)\,(1 - p'_{ng}) \tag{E2j}$$

$$K_g = I_g\,\sigma'_{LW}\,(1 - \overline{\epsilon}_v)\,p'_{ng} \tag{E2k}$$

$$L_g = I_g\,(1 - \sigma'_{LW}) \tag{E2l}$$

$$p'_{ng} = p_{ng}\,(1 - p_{n\alpha}) \tag{E2m}$$





and the equations for the snow-covered under-story fraction are

$$A_n = LW \downarrow p_{ng} \tag{E3a}$$

$$B_n = A_n \, \sigma_{fLW} \, (1 - \epsilon_v) \tag{E3b}$$

$$C_n = A_n \, (1 - \sigma_{fLW}) \tag{E3c}$$

$$D_n = C_n \, (1 - \epsilon_n) \tag{E3d}$$

$$E_n = D_n \, (1 - \sigma'_{LW}) \tag{E3e}$$

$$F_n = \overline{\sigma}_{fLW} \, \sigma \, \epsilon_v \, T_v^4 \, p_{ng} \tag{E3f}$$

$$G_n = F_n \, (1 - \epsilon_n) \tag{E3g}$$

$$H_n = G_n \, (1 - \sigma'_{LW}) \tag{E3h}$$

$$I_n = \sigma \, \epsilon_n \, T_n^4 \, p_{ng} \tag{E3i}$$

$$J_n = I_n \, \sigma'_{LW} \, (1 - \epsilon_v) \, (1 - p''_{ng}) \tag{E3j}$$

$$K_n = I_n \, \sigma'_{LW} \, (1 - \epsilon_v) \, p''_{ng} \tag{E3k}$$

$$L_n = I_n \, (1 - \sigma'_{LW}) \tag{E3l}$$

$$p''_{ng} = p_{ng} + p_{n\alpha} \, (1 - p_{ng}) \tag{E3m}$$

where the different terms are indicated in Fig. 4. In MEB, the ground-based snowpack depth can increase to the point that it buries the canopy, thus for both the snow-covered and snow free under-story fractions a modified snow fraction is defined as

$$\sigma'_{LW} = (1 - p'_{ng}) \, \sigma_{LW} + p'_{ng} \, \sigma_{fLW} \tag{E4}$$

The factor, $\sigma_{fLW}$, over the understory snow-covered fraction of the grid box is modeled quite simply
from Eq. 46. The net longwave radiation for the under-story, snowpack and vegetation canopy are therefore defined, respectively, as

$$LW_{netg} = C_g + F_g + J_g + J_n - D_g - G_g - I_g \tag{E5a}$$

$$LW_{netn} = C_n + F_n + K_n + K_g - D_n - G_n - I_n \tag{E5b}$$

$$
\begin{aligned}
LW_{netv} = {} & A_g + D_g + G_g + I_g + A_n + D_n + G_n + I_n \\
& - B_g - C_g - E_g - H_g - 2F_g - J_g - L_g - K_g
\end{aligned}
$$

$$\qquad\qquad - B_n - C_n - E_n - H_n - 2F_n - J_n - L_n - K_n \tag{E5c}$$

where the upwelling longwave radiation is computed from

$$LW \uparrow = LW \downarrow - LW_{netg} - LW_{netn} - LW_{netv} \tag{E6}$$

The inclusion of the snow-buried canopy fraction in Eq.s E2m and E3m causes all of the vegetation
transmission and below canopy fluxes to vanish as $p_{ng}$ and $p_{n\alpha} \to 0$ so that the only longwave radiative exchanges occur between the atmosphere and the snowpack in this limit.





**E1    Net Longwave radiation flux derrivatives**

The first order derivatives of the net longwave radiation terms are needed in order to solve the system
of linearized surface energy budget equations (Eq.s I1-I3). The Taylor series expansion (neglecting
higher order terms) is expressed as

$$LW_{net\,i}^{+} = LW_{net\,i} + \sum_{j=1}^{N_{seb}} \frac{\partial L_{net\,i}}{\partial T_j}\left(T_j^{+} - T_j\right) \qquad (i = 1, N_{seb}) \qquad (E7)$$

where $N_{seb}$ represents the number of surface energy budgets, and $i$ and $j$ represent the indexes for
each energy budget. The superscript $+$ represents the variable at time $t + \Delta t$, while by default, no
superscript represents the value at time $t$. Eq. E7 therefore results in a $N_{seb} \, x \, N_{seb}$ Jacobian matrix
(3x3 for MEB). The matrix coefficients are expressed as

$$\frac{\partial LW_{net\,v}}{\partial T_v} = \frac{\partial G_g}{\partial T_v} - \frac{\partial H_g}{\partial T_v} - 2\frac{\partial F_g}{\partial T_v} + \frac{\partial G_n}{\partial T_v} - \frac{\partial H_n}{\partial T_v} - 2\frac{\partial F_n}{\partial T_v} \qquad (E8a)$$

$$\frac{\partial LW_{net\,v}}{\partial T_g} = \frac{\partial I_g}{\partial T_g} - \frac{\partial J_g}{\partial T_g} - \frac{\partial K_g}{\partial T_g} - \frac{\partial L_g}{\partial T_g} \qquad (E8b)$$

$$\frac{\partial LW_{net\,v}}{\partial T_n} = \frac{\partial I_n}{\partial T_n} - \frac{\partial J_n}{\partial T_n} - \frac{\partial K_n}{\partial T_n} - \frac{\partial L_n}{\partial T_n} \qquad (E8c)$$

$$\frac{\partial LW_{net\,g}}{\partial T_v} = \frac{\partial F_g}{\partial T_v} - \frac{\partial G_g}{\partial T_v} \qquad (E8d)$$

$$\frac{\partial LW_{net\,g}}{\partial T_g} = \frac{\partial J_g}{\partial T_g} - \frac{\partial I_g}{\partial T_g} \qquad (E8e)$$

$$\frac{\partial LW_{net\,g}}{\partial T_n} = \frac{\partial J_n}{\partial T_n} \qquad (E8f)$$

$$\frac{\partial LW_{net\,n}}{\partial T_v} = \frac{\partial F_n}{\partial T_v} - \frac{\partial G_n}{\partial T_v} \qquad (E8g)$$

$$\frac{\partial LW_{net\,n}}{\partial T_g} = \frac{\partial K_g}{\partial T_g} \qquad (E8h)$$

$$\frac{\partial LW_{net\,n}}{\partial T_n} = \frac{\partial J_n}{\partial T_n} - \frac{\partial I_n}{\partial T_n} \qquad (E8i)$$





Using Eq. E5 to evaluate the derivatives we have

$$\frac{\partial LW_{net\,v}}{\partial T_v} = \frac{4}{T_v}\left(G_g - H_g - 2F_g + G_n - H_n - 2F_n\right) \tag{E9a}$$

$$\frac{\partial LW_{net\,v}}{\partial T_g} = \frac{4}{T_g}\left(I_g - J_g - K_g - L_g\right) \tag{E9b}$$

$$\frac{\partial LW_{net\,v}}{\partial T_n} = \frac{4}{T_n}\left(I_n - J_n - K_n - L_n\right) \tag{E9c}$$

$$\frac{\partial LW_{net\,g}}{\partial T_v} = \frac{4}{T_v}\left(F_g - G_g\right) \tag{E9d}$$

$$\frac{\partial LW_{net\,g}}{\partial T_g} = \frac{4}{T_g}\left(J_g - I_g\right) \tag{E9e}$$

$$\frac{\partial LW_{net\,g}}{\partial T_n} = \frac{4}{T_n}J_n \tag{E9f}$$

$$\frac{\partial LW_{net\,n}}{\partial T_v} = \frac{4}{T_v}\left(F_n - G_n\right) \tag{E9g}$$

$$\frac{\partial LW_{net\,n}}{\partial T_g} = \frac{4}{T_g}K_g \tag{E9h}$$

$$\frac{\partial LW_{net\,n}}{\partial T_n} = \frac{4}{T_n}\left(J_n - I_n\right) \tag{E9i}$$

so that from a coding perspective, the computation of the derivatives is trivial (using already computed quantities).

**Appendix F: Halstead coefficient maximum**

A maximum Halstead coefficient is imposed by estimating which value of $\delta_v$ that is needed to just evaporate any existing intercepted water, $W_{rv}$, given the conditions at the beginning of the time step. Assuming that phase changes are small, and neglecting canopy drip and any condensation from transpiration, the time-differenced prognostic equation for intercepted water on canopy vegetation (Eq. 68) can be approximated as:

$$\frac{W_{rv}^{+} - W_{rv}}{\Delta t} = (1 - \chi_v)(1 - p_{ng}p_{\alpha n})P_r - E_r \tag{F1}$$

Assuming that all existing water evaporates in one time step (i.e. $W_{rv}^{+} = 0$), and substituting the full expression for $E_r$ (Eq. C5) into Eq. F1, the maximum value of $\delta_v$ can be determined as

$$\delta_{v,max} = \frac{\left[(1 - \chi_v)\,(1 - p_{ng}p_{\alpha n})\,P_r + (W_{rv}/\Delta t)\right](L/L_v)}{\rho_a\,(1 - p_{nv})\,k_v\left\{\left[p_{ng}\,(1 - p_{\alpha n})\,/R_{avn-c}\right] + \left[(1 - p_{ng})\,/R_{avg-c}\right]\right\}(q_{sat\,v} - q_c)} \tag{F2}$$

Eq. F2 is an approximation since all of the variables on the RHS use conditions from the start of the time step, however, this method has proven to greatly reduce the risk for occasional numerical artifacts (jumps) and the associated need for mass corrections (if net losses in mass exceed the updated test value for interception storage).





**Appendix G: Energy and Mass conservation**

**G1  Energy Conservation**

The soil and snowpack prognostic temperature equations can be written in flux form for $k = 1, N_g$ soil layers and $k = 1, N_n$ snow layers as

$$\mathcal{C}_{g,k}\frac{\partial T_{g,k}}{\partial t} = G_{g,k-1} - G_{g,k} + L_f\,\Phi_{g,k} \tag{G1}$$

$$\mathcal{C}_{n,k}\frac{\partial T_{n,k}}{\partial t} = G_{n,k-1} - G_{n,k} + L_f\,\Phi_{n,k} + \xi_{n,k-1} - \xi_{n,k} + SW_{nn}\left(\tau_{n,k-1} - \tau_{n,k}\right) \tag{G2}$$

The total energy balance of the vegetation canopy-soil-snowpack system is conserved at each time step, $\Delta t$, and can be obtained by summing the discrete time forms of Eq. 4, Eq. G1, and Eq. G2 for all soil, snow and the single bulk vegetation layers yielding

$$\mathcal{C}_v\Delta T_v + \sum_{k=1}^{N_g}\mathcal{C}_{g,k}\,\Delta T_{g,k} + p_n\sum_{k=1}^{N_n}\mathcal{C}_{n,k}\,\Delta T_{n,k} =$$
$$\Delta t\left[G_{g,0} + p_n\,G_{n,0} + R_{nv} - H_{v-c} - LE_{v-c} + L_f\left(\Phi_v + \sum_{k=1}^{N_g}\Phi_{g,k} + \sum_{k=1}^{N_n}\Phi_{n,k}\right)\right] \tag{G3}$$

where $\Delta T_x = T_x(t + \Delta t) - T_x(t)$. The surface boundary conditions for Eq. 4 and Eq. 6 are, respectively,

$$G_{g,0} = (1 - p_n)\left(R_{ng} - H_g - LE_g\right) + p_n\left(G_{gn} + \tau_{n,N_n}SW_{nn}\right) \tag{G4}$$

$$G_{n,0} = R_{nn} - H_n - LE_n - H_n - LE_{n-N} \tag{G5}$$

$$\tau_{n,0} = 1 \tag{G6}$$

$$\xi_{n,0} = 0 \tag{G7}$$

Eq. G6 signifies that the net shortwave radiation at the surface enters the snowpack, and Eq. G7 represents the fact that energy changes owing to the time evolving snow grid can only arise in the surface layer owing to exchanges with the sub-surface layer. Snowfall is assumed to have the same tempera-

ture as the snowpack, thus a corresponding cooling/heating term does not appear in Eq. G5, although the corresponding mass increase must appear in the snow water budget equation (see Sect. 2.7).

The lower boundary conditions for Eq. G1 and Eq. G2 are, respectively,

$$G_{g,N_g} = 0 \tag{G8}$$

$$\xi_{n,N_n} = 0 \tag{G9}$$

The appearance of the same discrete form for $\Phi$ in both the energy and mass budget equations ensures enthalpy conservation. Owing to Eq.s G7 and G9, the total effective heating of the snowpack owing to grid adjustments is

$$\int_0^{D_{Nn}} \xi_n\,dD_n \;=\; 0 \tag{G10}$$





where $D_{Nn}$ represents the total snow depth. Thus this term only represents a contribution from contiguous snow layers, not from a source external to the snowpack. The energy storage of the snow-soil-vegetation system is balanced by the net surface radiative and turbulent fluxes and internal phase changes (solid and liquid phases of water substance).

**G2  Mass Conservation**

The soil and snowpack prognostic mass equations can be written in flux form for $k = 2, N_{gw}$ soil layers and $k = 1, N_n$ snow layers as

$$p_{ng} \frac{\partial W_{n,k}}{\partial t} = p_{ng} \left( F_{nl,k-1} - F_{nl,k} - \Phi_{n,k} + \xi_{nl,k} - \xi_{nl,k-1} \right) \qquad (k = 2, N_n) \qquad (G11)$$

$$\rho_w \Delta z_{g,1} \frac{\partial w_{g,k}}{\partial t} = F_{g,k-1} - F_{g,k} - \Phi_{g,k} - \mathcal{F}_{2,k} \max(0, E_{tr}) \qquad (k = 2, N_{gw}) \qquad (G12)$$

$$\rho_w \Delta z_{g,1} \frac{\partial w_{gf,k}}{\partial t} = \Phi_{g,k} \qquad (k = 2, N_{gw}) \qquad (G13)$$

The total grid-box water budget at each time step is obtained by summing the budget equations for the surface layers (Eq.s 68-72) together with those for the sub-surface layers (Eq.s G11-G13) to have

$$\Delta W_r + \Delta W_{rn} + p_{ng} \sum_{k=1}^{N_n} \Delta W_{n,k} + \rho_w \sum_{k=1}^{N_{gw}} \Delta z_{g,k} (w_{gk} + w_{gfk}) =$$
$$\Delta t \Big[ P_r + P_s - R_0 - F_{g,N_{gw}} - (1 - p_{ng}) E_g - E_v - p_{ng} E_n \qquad (G14)$$
$$- \Phi_v - \sum_{k=1}^{N_g} \Phi_{g,k} - \sum_{k=1}^{N_n} \Phi_{n,k} \Big]$$

where $R_0$ can simply be a diagnostic or coupled with a river routing scheme (Habets et al., 2008; Decharme et al., 2012; Getirana et al., 2015). The soil water lower boundary condition, $F_{g,N_{gw}}$ represents the so-called base-flow or drainage leaving the lowest hydrological layer which can then be transfered as input to a river routing scheme (see references above) or to a ground water scheme. In such instances, it can be negative if an option to permit a ground water inflow is activated (Vergnes et al., 2014). The soil liquid water and equivalent frozen water equivalent volumetric water content extend down to layer $N_{gw}$, where $N_{gw} \leq N_g$. Note that the vertical soil water transfer or evolution is not computed below $z_g (k = N_{gw})$, whereas heat transfer can be. In order to compute the thermal properties for deep soil temperature (thermal conductivity and heat capacity for example), soil moisture estimates are needed: values from the soil are extrapolated downward assuming hydrostatic equilibrium A detailed description of the soil model is given by Decharme et al. (2011) and Decharme et al. (2013).

Note that Eq. G11 is snow-relative, therefore this equation must be multiplied by the ground-based snow fraction, $p_{ng}$, to be grid box relative for coupling with the soil and vegetation water storage terms. The lower boundary condition for liquid water flow, $F_{nl,N_n}$, is defined as the liquid water exceeding the lowest maximum snow layer liquid water holding capacity. $\xi_{nl}$ represents the internal mass changes of a snowpack layer when the vertical grid is reset. When integrated over the entire snowpack depth, this term vanishes (analogous to Eq. G10 for the snowpack tempera-





ture equation). See Boone and Etchevers (2001) and Decharme et al. (2016) for details on the snow model processes.

The equations describing flooding are not described in detail here as this parameterization is independent of MEB, and it is described in detail by Decharme et al. (2012). The coupling of MEB with the interactive flooding scheme will be the subject of a future paper.

### Appendix H: Implicit numerical coupling with the atmosphere

The land-atmosphere coupling is accomplished through the atmospheric model vertical diffusion
(heat, mass, momentum, chemical species, aerosols, etc.) and radiative schemes. Owing to the potential for relatively large diffusivity, especially in the lower atmosphere near the surface, fairly strict time step constraints must be applied. In this section, a fully implicit time scheme (with an option for explicit coupling) is described. There are two reasons for using this approach: i) an implicit coupling is more numerically stable, especially for time steps typical of GCM applications, but also
for some NWP models, and ii) the methodology permits code modularity in that the land surface model routines can be independent of the atmospheric model code and they can be called using a standard interface, which is the philosophy of SURFEX (Masson et al., 2013). The coupling follows the methodology first proposed by Polcher et al. (1998) which was further generalized by Best et al. (2004).

The atmospheric turbulence scheme is generally expressed as a second order diffusion equation in the vertical (which is assumed herein) and it is discretized using the backward difference time scheme. Note that a semi-implicit scheme, such as the Crank-Nicolson (Crank and Nicolson, 1947), could also be used within this framework. thus the equations can be cast as a tri-diagonal matrix. Assuming a fixed for zero (the general case) upper boundary condition at the top of the atmosphere,
the diffusion equations for the generic variable $\phi$ can be cast as a linear function of the variable in the layer below (Richtmeyer and Morton, 1967) as

$$\phi_k^+ = B_{\phi,k} + A_{\phi,k}\,\phi_{k+1}^+ \qquad\qquad (k = 1, N_a - 1) \qquad\qquad\qquad \text{(H1)}$$

where $N_a$ represents the number of atmospheric model layers, $k = 1$ represents the uppermost layer with $k$ increasing with decreasing height above the surface, and the superscript $+$ indicates the value
of $\phi$ at time $t + \Delta t$ (at the end of the time step). The coefficients $A_{\phi,k}$ and $B_{\phi,k}$ are computed in a downward sweep within the turbulence scheme and thus consist in atmospheric prognostic variables, diffusivity, heat capacities and additional source terms from layer $k$ and above evaluated at time level $t$ (Polcher et al., 1998). As shown by Best et al. (2004), the equation for the lowest atmospheric model layer can be expressed using a flux lower boundary condition as

$$\phi_{N_a}^+ = B_{\phi,N_a} + A_{\phi,N_a}\,F_{\phi,N_a+1}^+ \qquad\qquad\qquad\qquad\qquad\qquad\qquad \text{(H2)}$$





where $F^+_{\phi,N_a+1}$ is the implicit surface flux from one or multiple surface energy budgets. Technically, only the $B_{\phi,N_a}$ and $A_{\phi,N_a}$ coefficients are needed by the LSM in order to compute the updated land surface fluxes and temperatures which are fully implicitly coupled with the atmosphere. Once $F^+_{\phi,N_a+1}$ has been computed by the LSM, it can be returned to the atmospheric turbulence

scheme which can then solve for $\phi^+_k$ from $k=N_a$ to $k=1$ (i.e. the upward sweep). For explicit land-atmosphere coupling or offline land-only applications, the coupling coefficients can be set to $A_{\phi,N_a}=0$ and $B_{\phi,N_a}=\phi_{N_a}$ in the driving code.





## Appendix I:  Numerical solution of the surface energy budgets

### I1   Discretization of surface energy budgets

The surface energy budget equations (Eq.s 4-6) are integrated in time using the implicit backward difference scheme. They can be written in discretized form as

$$
\mathcal{C}_v \frac{(T_v^+ - T_v)}{\Delta t} = \frac{\partial LW_{netv}}{\partial T_v}(T_v^+ - T_v) + \frac{\partial LW_{netv}}{\partial T_{g,1}}(T_{g,1}^+ - T_{g,1})
$$
$$
+ \frac{\partial LW_{netv}}{\partial T_{n,1}}(T_{n,1}^+ - T_{n,1}) + SW_{netv} + LW_{netv}
$$
$$
+ \varphi_v \left( \mathcal{A}_v T_v^+ - \mathcal{A}_c T_c^+ \right)
$$
$$
+ h_{sv}\varphi_v L \left[ q_{satv} + \frac{\partial q_{satv}}{\partial T_v}(T_v^+ - T_v) - q_c^+ \right] \tag{I1}
$$

$$
\mathcal{C}_{g,1} \frac{(T_{g,1}^+ - T_{g,1})}{\Delta t} = \left[ \frac{\partial LW_{netg}}{\partial T_v}(T_v^+ - T_v) + \frac{\partial LW_{netg}}{\partial T_{g,1}}(T_{g,1}^+ - T_{g,1}) \right.
$$
$$
+ \frac{\partial LW_{netg}}{\partial T_{n,1}}(T_{n,1}^+ - T_{n,1}) + SW_{netg} + LW_{netg}
$$
$$
+ \varphi_g \left( \mathcal{A}_g T_g^+ - \mathcal{A}_c T_c^+ \right)
$$
$$
+ \varphi_g L \left\{ h_{sg}\left[ q_{satg} + \frac{\partial q_{satg}}{\partial T_g}(T_g^+ - T_g) \right] - h_a q_c^+ \right\}
$$

$$
\left. \right] (1 - p_{ng}) + p_{ng}\Lambda_{g,n}(T_{n,N_n}^* - T_{g,1}^+) - \Lambda_{g,1}(T_{g,1}^+ - T_{g,2}^+) \tag{I2}
$$

$$
p_{ng}\mathcal{C}_{n,1} \frac{(T_{n,1}^+ - T_{n,1})}{\Delta t} = \left\{ \frac{\partial LW_{netn}}{\partial T_v}(T_v^+ - T_v) + \frac{\partial LW_{netn}}{\partial T_{g,1}}(T_{g,1}^+ - T_{g,1}) \right.
$$
$$
+ \frac{\partial LW_{netn}}{\partial T_{n,1}}(T_{n,1}^+ - T_{n,1}) + SW_{netn} + LW_{netn}
$$
$$
+ (1 - p_{n\alpha})\varphi_{n-c}\left( \mathcal{A}_n T_n^+ - \mathcal{A}_c T_c^+ \right)
$$
$$
+ p_{n\alpha}\varphi_{n-a}\left( \mathcal{B}_n - \mathcal{B}_a + \mathcal{A}_n T_n^+ - \mathcal{A}_a T_a^+ \right)
$$
$$
+ (1 - p_{n\alpha})\varphi_{n-c}L_s\left[ q_{satin} + \frac{\partial q_{satin}}{\partial T_n}(T_n^+ - T_c^+) - q_c^+ \right]
$$
$$
+ p_{n\alpha}\varphi_{n-a}L_s\left[ q_{satin} + \frac{\partial q_{satin}}{\partial T_n}(T_n^+ - T_a^+) - q_a^+ \right]
$$
$$
\left. - \Lambda_{g,1}(T_{n,1}^+ - T_{n,2}^+) \right\} p_{ng} \tag{I3}
$$

The $q_{satx}^+$ and longwave radiation terms have been linearized with respect to $T_x$ (the longwave
radiation derivatives are given by Eq. E9). The superscript + corresponds to the values of variables at time $t + \Delta t$, while the absence of a superscript indicates variables evaluated at time $t$. Note that we have defined $\varphi_x = \rho_a / R_{ax}$ (kg m$^{-2}$ s$^{-1}$) for simplicity. The thermodynamic variable, $\mathcal{T}_x$, in the sensible heat flux terms have been expressed as a function of $T_x$ using Eq. 15. Several of the $\mathcal{B}_x$



terms have canceled out in the sensible heat flux terms in Eq.s I1-I3 since they are defined such that $\mathcal{B}_c = \mathcal{B}_v = \mathcal{B}_g = \mathcal{B}_n$. Note that compared to Eq.s 4-6, the phase change terms ($\Phi_x$) do not appear in Eq.s I1-I3. This is because they are evaluated as an adjustment after the energy budget and the fluxes have been computed.

In Eq. I2, $T^*_{n,N_n}$ represents a test temperature for the lowest snowpack layer. It is first computed using an implicit calculation of the combined snow-soil layers to get a first estimate of the snow-ground heat conduction inter-facial flux when simultaneously solving the surface energy budgets. The final snow temperature in this layer, $T^+_{n,N_n}$, is computed afterwards within the snow scheme: any difference between the resulting conduction flux and the test-flux in Eq. I2 is added to the soil as a correction at the end of the time step in order to conserve energy. In practice, this correction is generally small, especially since the snow fraction goes to unity very rapidly (i.e. for a fairly thin snowpack when using MEB; see Eq. 1). Thus, in this general case, the difference between the test flux and the final flux arise only owing to updates to snow properties within the snow scheme during the time step. Since $T^*_{n,N_n}$ is computed using an implicit solution method for the entire soil-snow continuum, it is also quite numerically stable. The use of a test flux permits a modular coupling between the snow scheme and the soil-vegetation parts of ISBA-MEB.

In order to solve Eq.s I1-I3 for the three unknown surface energy budget temperatures, $T^+_v$, $T^+_{g,1}$, and $T^+_{n,1}$, equations for the six additional unknowns, $T^+_a$, $T^+_c$, $q^+_a$, $q^+_c$, $T^+_{g,2}$ and $T^+_{n,2}$, must be defined. They can be expressed as linear equations in terms of $T^+_v$, $T^+_{g,1}$, and $T^+_{n,1}$, and their derivations are presented in the remaining sections of this Appendix.

### I2 Atmospheric temperature and specific humidity

The first step in solving the surface energy budget is to eliminate the lowest atmospheric energy and water vapor variables from the snow surface energy budget equation. They will also be used to diagnose the final flux exchanges between the canopy air space and overlying atmosphere.

From Eq. H2, the thermodynamic variable of the lowest atmospheric model variable at time $t + \Delta t$ is defined as

$$\mathcal{T}^+_{N_a} = B_{\mathcal{T},N_a} + A_{\mathcal{T},N_a} H^+ \tag{I4}$$

Note that using Eq. 15, we can rewrite Eq. I4 in terms of air temperature as

$$T_a{}^+ = B_{T_a} + A_{T_a} H^+ \tag{I5}$$

where $B_{T_a} = (B_{\mathcal{T},N_a} - \mathcal{B}_a)/\mathcal{A}_a$, $A_{T_a} = A_{\mathcal{T},N_a}/\mathcal{A}_a$, and $T_a$ is shorthand for $T(k = N_a)$. Substitution of Eq. 14 for $H$ in Eq. I5 and solving for $T^+_a$ yields

$$T^+_a = \mathscr{B}_{T_a} + \mathscr{A}_{T_a} T^+_c + \mathscr{C}_{T_a} T^+_n \tag{I6}$$



where

$$C = \mathcal{A}_a \left\{ 1 + A_{T_a} \left[ \varphi_{c-a} \left( 1 - p_{ng} p_{\alpha n} \right) + p_{ng} p_{\alpha n} \varphi_{n-a} \right] \right\} \tag{I7a}$$

$$\mathscr{A}_{T_a} = A_{T_a} \varphi_{c-a} \mathcal{A}_c \left( 1 - p_{ng} p_{\alpha n} \right) / C \tag{I7b}$$

$$\mathscr{B}_{T_a} = \left\{ B_{T_a} - \mathcal{B}_a + A_{T_a} \left[ \left( 1 - p_{ng} p_{\alpha n} \right) \varphi_{c-a} \left( \mathcal{B}_c - \mathcal{B}_a \right) + \right. \right.$$
$$\left. \left. p_{ng} p_{\alpha n} \varphi_{n-a} \left( \mathcal{B}_c - \mathcal{B}_a \right) \right] \right\} / C \tag{I7c}$$

$$\mathscr{C}_{T_a} = A_{T_a} p_{ng} p_{\alpha n} \varphi_{n-a} \mathcal{A}_c / C \tag{I7d}$$

$$\tag{I7e}$$

In analogous fashion to determining the air temperature, the specific humidity of the lowest atmospheric model variable at time $t + \Delta t$ is defined from Eq. H2 as

$$q_a^+ = B_{q,a} + A_{q,a} E^+ \tag{I8}$$

where again the subscript $q, a$ represents the values of the coefficients $A$ and $B$ for the lowest atmospheric model layer ($k = N_a$). Substitution of Eq. 21 for $E$ in Eq. I8 and solving for $T_a^+$ yields

$$q_a^+ = \mathscr{B}_{q,a} + \mathscr{A}_{q,a} q_c^+ + \mathscr{C}_{q,a} q_{satin}^+ \tag{I9}$$

where the coefficients are defined as

$$C = 1 + A_{q,a} \left[ \left( 1 - p_{ng} p_{\alpha n} \right) \varphi_{c-a} + \varphi_{n-a} h_{sn} p_{\alpha n} p_{ng} \right] \tag{I10a}$$

$$\mathscr{A}_{q,a} = A_{q,a} \varphi_{c-a} \left( 1 - p_{ng} p_{\alpha n} \right) / C \tag{I10b}$$

$$\mathscr{B}_{q,a} = B_{q,a} / C \tag{I10c}$$

$$\mathscr{C}_{q,a} = A_{q,a} \varphi_{n-a} h_{sn} p_{\alpha n} p_{ng} / C \tag{I10d}$$

### I3 Canopy air temperature and specific humidity

In order to close the energy budgets, $T_c^+$ and $q_c^+$ must be determined.

Assuming conservation of the heat flux between the different surfaces and the canopy air space, we have

$$\left( 1 - p_{ng} p_{n\alpha} \right) H_c^+ = p_{ng} \left( 1 - p_{n\alpha} \right) H_{n-c}^+ + \left( 1 - p_{ng} \right) H_g^+ + H_v^+ \tag{I11}$$

which can be expanded as

$$\varphi_{c-a} \left( 1 - p_{ng} p_{\alpha n} \right) \times$$
$$\left( \mathcal{B}_c + \mathcal{A}_c T_c^+ - \mathcal{B}_a - \mathcal{A}_a T_a^+ \right) = \mathcal{A}_c \left[ \varphi_g \left( T_g^+ - T_c^+ \right) \left( 1 - p_{ng} \right) + \varphi_v \left( T_v^+ - T_c^+ \right) \right.$$
$$\left. \varphi_{n-c} \left( T_n^+ - T_c^+ \right) p_{ng} \left( 1 - p_{\alpha n} \right) \right] \tag{I12}$$





Note that the above conservation equation does not include the part of the snow sensible heat flux which is in direct contact with the atmosphere ($H_{n-a}$) since it was already accounted for in the expression for $T_a^+$ via Eq. I5. Eliminating $T_a^+$ using Eq. I6 and solving for $T_c^+$ yields

$$T_c^+ = a_{Tc} + b_{Tc} T_v^+ + c_{Tc} T_g^+ + d_{Tc} T_n^+ \tag{I13}$$

with the coefficients

$$C = \varphi_{c-a}\left(1 - p_{ng}\,p_{\alpha n}\right)\left(\mathcal{A}_c - \mathcal{A}_a\,\mathscr{A}_{Ta}\right) + \qquad$$
$$\mathcal{A}_c\left[\varphi_v + \varphi_g\left(1 - p_{ng}\right) + \varphi_{n-c}\,p_{ng}\left(1 - p_{\alpha n}\right)\right] \tag{I14a}$$

$$a_{Tc} = \left[\varphi_{c-a}\left(1 - p_{ng}\,p_{\alpha n}\right)\left(\mathcal{B}_a - \mathcal{B}_c + \mathcal{A}_a\,\mathscr{B}_{Ta}\right)\right]/C \tag{I14b}$$

$$b_{Tc} = \mathcal{A}_c\,\varphi_v/C \tag{I14c}$$

$$c_{Tc} = \mathcal{A}_c\,\varphi_g\left(1 - p_{ng}\right)/C \tag{I14d}$$

$$d_{Tc} = \left[\mathcal{A}_c\,\varphi_{n-c}\,p_{ng}\left(1 - p_{\alpha n}\right) + \mathcal{A}_a\,\mathscr{C}_{Ta}\,\varphi_{c-a}\left(1 - p_{ng}\,p_{\alpha n}\right)\right]/C \tag{I14e}$$

In an analogous fashion for canopy air temperature determination, assuming conservation of the vapor flux between the different surfaces and the canopy air space,

$$\left(1 - p_{ng} p_{n\alpha}\right) E_c^+ \;=\; p_{ng}\left(1 - p_{n\alpha}\right) E_{n-c}^+ + \left(1 - p_{ng}\right) E_g^+ + E_v^+ \tag{I15}$$

which can be expanded using the definitions of the evaporative fluxes, $E_x$, from Eq.s 17-I15 together with the definitions of $q_g$ from Eq. 22 and $q_a^+$ from Eq. I9 as

$$\varphi_{c-a}\left(1 - p_{ng} p_{\alpha n}\right) \times$$
$$\left[q_c^+\left(1 - \mathscr{A}_{q,a}\right) - \mathcal{B}_{q,a} - \mathscr{C}_{q,a}\,q_{sat\,in}^+\right] = \left[\varphi_g\left(h_{sg}\,q_{sat\,g}^+ - h_a\,q_c^+\right)\left(1 - p_{ng}\right) + \varphi_v\,h_{sv}\left(q_{sat\,v}^+ - q_c^+\right)\right.$$
$$\left. \varphi_{n-c}\,h_{sn}\left(q_{sat\,in}^+ - q_c^+\right) p_{ng}\left(1 - p_{\alpha n}\right)\right] \tag{I16}$$

Owing to the linearization of the $q_{sat\,x}$ terms about $T_x$, Eq. I16 can be solved for $q_c^+$ as a function of the surface energy budget temperatures as

$$q_c^+ = a_{qc} + b_{qc} T_v^+ + c_{qc} T_g^+ + d_{qc} T_n^+ \tag{I17}$$





where the coefficients are defined as

$$
\begin{aligned}
C =& \varphi_{c-a}\left(1 - p_{ng}\,p_{n\alpha}\right)\left(1 - \mathscr{A}_{q,a}\right) + \varphi_g\,h_N\left(1 - p_{ng}\right) \\
& + \varphi_v\,h_{sv} + \varphi_{n-c}\,h_{sn}\,p_{ng}\left(1 - p_{n\alpha}\right)
\end{aligned}
\tag{I18a}
$$

$$
\begin{aligned}
a_{qc} =& \Big\{ \left(1 - p_{ng}\,p_{n\alpha}\right)\varphi_{c-a}\,\mathscr{B}_{q,a} + \varphi_v\,h_{sv}\left(q_{sat\,v} - \frac{\partial q_{sat\,v}}{\partial T_v}T_v\right) \\
& + \varphi_g\,h_{sg}\left(q_{sat\,g} - \frac{\partial q_{sat\,g}}{\partial T_g}T_g\right)\left(1 - p_{ng}\right) \\
& + \varphi_{n-c}\,h_{sn}\left(q_{sati\,n} - \frac{\partial q_{sati\,n}}{\partial T_n}T_n\right)p_{ng}\left(1 - p_{n\alpha}\right)\Big\}/C
\end{aligned}
\tag{I18b}
$$

$$
b_{qc} = h_{sv}\,\varphi_v\,\frac{\partial q_{sat\,v}}{\partial T_v}/C
\tag{I18c}
$$

$$
c_{qc} = h_{sg}\,\varphi_g\,\frac{\partial q_{sat\,g}}{\partial T_g}\left(1 - p_{ng}\right)/C
\tag{I18d}
$$

$$
d_{qc} = h_{sn}\,\varphi_{n-c}\,\frac{\partial q_{sati\,n}}{\partial T_n}\,p_{ng}\left(1 - p_{n\alpha}\right)/C
\tag{I18e}
$$

### I4  Sub-surface temperatures

The sub-surface conduction heat fluxes (Eq.s 47-49) can be expressed in compact form as

$$
G^+_{x,k} = \Lambda_{x,k}\left(T^+_{x,k} - T^+_{x,k+1}\right)
\tag{I19}
$$

where $\Lambda_{x,k}$ represents the ratio of the inter-facial thermal conductivity to the thickness between the mid-points of contiguous layers ($k$ and $k + 1$). Using the methodology described in Appendix H for the atmospheric diffusion scheme, the soil and snow heat diffusion equation (both using the form of Eq. G1) can be defined in an analogous fashion as

$$
T^+_{g,k} = B_{g,k} + A_{g,k}\,T^+_{g,k-1} \qquad (k = 2, N_g)
\tag{I20}
$$

where the coefficients $B_{g,k}$ and $A_{g,k}$ are determined during the upward sweep (first step of the tridiagonal solution) from the base of the soil to the sub-surface soil and snow layers as described by Richtmeyer and Morton (1967). The resulting coefficients for the soil are defined as

$$
C = \left(\mathcal{C}_{g\,k}/\Delta t\right) + \Lambda_{g\,k-1} + \Lambda_{g\,k}\left(1 - A_{g\,k+1}\right)
\tag{I21a}
$$

$$
B_{g\,i} = \left[\left(\mathcal{C}_{g\,k}/\Delta t\right)T_{g\,k} + \Lambda_{g\,k}\,B_{g\,k+1}\right]/C \qquad (2 \leq k \leq N_g - 1)
\tag{I21b}
$$

$$
A_{g\,k} = \Lambda_{g\,k-1}/C
\tag{I21c}
$$

The same form holds for the snow layers. The upward sweep is performed before the evaluation of the energy budget, thus Eq. I20 is used to eliminate $T^+_{g,2}$ and $T^+_{n,2}$ from Eq.s I2 and I3, respectively. To do this, the sub-surface implicit fluxes in Eq.s 5 and 6 can be expressed, respectively, as

$$
G^+_{g,1} = \Lambda_{g,1}\left[T^+_{g,1}\left(1 - A_{g,2}\right) + B_{g,2}\right]
\tag{I22a}
$$

$$
G^+_{n,1} = \Lambda_{n,1}\left[T^+_{n,1}\left(1 - A_{n,2}\right) + B_{n,2}\right]
\tag{I22b}
$$



**I5 Surface stresses**

Using the same surface-atmosphere coupling methodology as for temperature and specific humidity, the u-wind component in the lowest atmospheric model layer can be expressed as

$$u_a^+ = B_{ua} + A_{ua}\tau_x^+ \tag{I23}$$

The surface $u$ component momentum exchange with the atmosphere is expressed as

$$\tau_x^+ = -u_a^+ \left[ (1 - p_{ng}p_{n\alpha})\varphi_{Dc-a} + p_n p_{n\alpha}\varphi_{Dn-a} \right] \tag{I24}$$

where it includes stresses from the snow-buried and non-snow buried portions of the surface consistent with the fluxes of heat and water vapor. For simplicity, we have defined

$$\varphi_{Dx} = \rho_a V_a C_{Dx} \tag{I25}$$

and $C_D$ is the surface drag coefficient which is defined following Noilhan and Mahfouf (1996). Eliminating $\tau_x^+$ from Eq. I24 using Eq. I25 gives

$$u_a^+ = \frac{B_{ua}}{1 + A_{ua}\varphi_{Dc}} \tag{I26}$$

where for convenience we have defined the average drag coefficient as

$$\varphi_{Dc} = (1 - p_{ng}p_{n\alpha})\varphi_{Dc-a} + p_{ng}p_{n\alpha}\varphi_{Dn-a} \tag{I27}$$

The net $u$-momentum flux from the surface to the canopy air space is expressed as

$$\tau_x^+ = -\frac{B_{ua}\varphi_{Dc}}{(1 + A_{ua}\varphi_{Dc})} \tag{I28}$$

Finally, the vector momentum flux in the atmosphere can be computed from the scalar friction velocity:

$$u^* = \left( \frac{\varphi_{Dc} V_a^+}{\rho_a} \right)^{1/2} \tag{I29}$$

where $V_a^+$ is the updated wind speed (computed from $u_a^+$ and $v_a^+$). Note that $v_a^+$ and $\tau_y^+$ are computed in the same manner, but using $B_{va}$ from the atmosphere (note that $A_{va} = A_{ua}$).

**I6 Summary: Final solution of the implicitly coupled equations**

The fully implicit solution of the surface and atmospheric variables proceeds for each model time step as follows:

1. Within the atmospheric model, perform the downward sweep of the tri-diagonal matrix within the turbulent diffusion scheme of the atmospheric model to obtain the $A_{\phi,k}$ and $B_{\phi,k}$ coefficients for each diffused variable ($\phi = T$, $q$, $u$, and $v$) for each layer of the atmosphere





$(k = 1, N_a)$. Update $\mathcal{A}_a$ and $\mathcal{B}_a$, then pass these values along with the aforementioned coupling coefficients at the lowest atmospheric model layer (i.e. $A_{T,a}$, $B_{T,a}$, $A_{q,a}$, $B_{q,a}$, $A_{u,a}$, $B_{u,a}$, and $B_{v,a}$) to the land surface model. These coefficients are then used to eliminate $T_a^+$ and $q_a^+$ from the implicit surface energy budget equations (Eq.s I1-I3).

2. Within the land surface model, perform the upward sweep of the tri-diagonal matrix within the soil and snow layers to determine the $A_{n,k}$, $B_{n,k}$, $A_{g,k}$, and $B_{g,k}$, coefficients for the soil and snow layers (from soil layer $N_g$ to layer 2, and again from soil layer $N_g$ to layer 2 of the snow scheme). Note that coefficients for layer 1 of the snow and soil schemes are not needed since they correspond to the linearized surface energy budgets (next step).

3. Within the land surface model, the expressions for $T_a^+$ (Eq. I6), $q_a^+$ (Eq. I9), $T_c^+$ (Eq. I13), $q_c^+$ (Eq. I17), $T_{g,2}^+$ (Eq. I22a)and $T_{n,2}^+$ (Eq. I22b) can now be substituted into the energy budget equations (Eq.s I1-I3) which can then be readily solved for $T_v^+$, $T_{g,1}^+$, and $T_{n,1}^+$.

4. Within the land surface model, perform back-substitution (using $T_{g,1}^+$ as the upper boundary condition) to obtain $T_{g,k}^+$ for soil layers $k = 2, N_g$ using Eq. I20.

5. Within the land surface model, call the explicit snow-process scheme to update the snow scheme temperature, $T_{n,k}^+$, and the snow mass variables for snow layers $k = 2, N_n$. The implicit snow surface fluxes, $R_{n,n}^+$, $H_n^+$ and $E_n^+$, are used as the upper boundary condition along with the implicit soil temperature, $T_{g,1}^+$, to compute the updated lower snowpack boundary condition (i.e. the snow-soil inter-facial flux, $G_{gn}$).

6. Within the land surface model, compute $V_a^+$ (See Section I5). Diagnose $T_a^+$, $T_c$+, $q_a^+$ and $q_c^+$ (again, using the equations mentioned in Step 3) in order to compute the updated (implicit) fluxes. The updated evapotranspiration (Eq.s C4-C8) and snow melt water mass fluxes are used within the hydrology schemes to update the different water storage variables for the soil and vegetation canopy (Eq.s 68-72).

7. Within the atmospheric model, perform back-substitution (using $H^+$, $E^+$, $\tau_x^+$ and $\tau_y^+$ as the lower boundary conditions: Eq. H2) to obtain updated profiles (or turbulent tendencies, depending on the setup of the atmospheric model) of $\mathcal{T}_k$, $q_k$, $u_k$ and $v_k$ for atmospheric layers $k = 1, N_a$. Finally, the updated upwelling shortwave, $SW \uparrow$, and implicit longwave flux, $LW \uparrow^+$ (or equivalently, the effective emissivity and implicit longwave radiative temperature, $T_{rad}^+$) are returned to the atmospheric model as lower boundary conditions for the respective radiative schemes.

Alternately, in offline mode, $A_{\phi,a} = 0$ and $B_{\phi,a} = \phi_a$ in the driving routine in Step 1, and the solution procedure ends at Step 6. Finally, if multiple patches and/or tiles are being used within the grid call of interest, the corresponding fractional-area weighted fluxes are passed to the atmospheric model in Step 7.





**Table 1.** Description of the patches for the natural land surface sub-grid tile. The values for the 19-class option are shown in the leftmost three columns, and those for the 12-class option are shown in the rightmost three columns (the name and description are only given if they differ from the 19-class values). MEB can currently be activated for the forest classes: 4-6 (for both the 12 and 19 class options), and 13-17.

| Index | Name | Description | Index | Name | Description |
|---|---|---|---|---|---|
| 1 | NO | Bare Soil | 1 | | |
| 2 | ROCK | Rock | 2 | | |
| 3 | SNOW | Permanent snow or ice | 3 | | |
| 4 | TEBD | Temperate broad leaf | 4 | TREE | Broad leaf |
| 5 | BONE | Boreal evergreen needle leaf | 5 | CONI | Evergreen needle leaf |
| 6 | TRBE | Tropical evergreen broad leaf | 6 | EVER | Evergreen broad leaf |
| 7 | C3 | C3 Crops | 7 | | |
| 8 | C4 | C4 Crops | 8 | | |
| 9 | IRR | Irrigated crops | 9 | | |
| 10 | GRAS | Temperate Grassland | 10 | | |
| 11 | TROG | Tropical grassland | 11 | | |
| 12 | PARK | Bog, park, garden | 12 | | |
| 13 | TRBD | Tropical broad leaf | | | |
| 14 | TEBE | Temperate evergreen broad leaf | | | |
| 15 | TENE | Temperate evergreen needle leaf | | | |
| 16 | BOBD | Boreal broad leaf | | | |
| 17 | BOND | Boreal needle leaf | | | |
| 18 | BOGR | Boreal grassland | | | |
| 19 | SHRB | Shrubs | | | |





**Table 2.** Surface vegetation canopy turbulence parameters which are constant.

| Symbol | Definition | Unit | Value | Reference | Comment |
|---|---|---|---|---|---|
| $a_{av}$ | canopy conductance scale factor | m s$^{-1/2}$ | 0.01 | Choudhury and Monteith (1988) | Eq. 26 |
| $\phi'_v$ | attenuation coeff. for wind | - | 3 | Choudhury and Monteith (1988) | p 386 |
| $lw$ | leaf width | m | 0.02 | | |
| $\phi_v$ | attenuation coeff. for mom. | - | 2 | Choudhury and Monteith (1988) | p 386 |
| $z_{0g}$ | roughness of soil surface | m | 0.007 | | |
| $\chi_L$ | Ross-Goudriaan leaf angle dist. | - | 0.12 | Monteith (1975) | p 26 |
| $u_l$ | Typical local wind speed | m s$^{-1}$ | 1 | Sellers et al. (1996) | Eq. B7 |
| $\upsilon$ | Kinematic viscos. of air | m$^2$ s$^{-1}$ | $0.15 \times 10^{-4}$ | | |





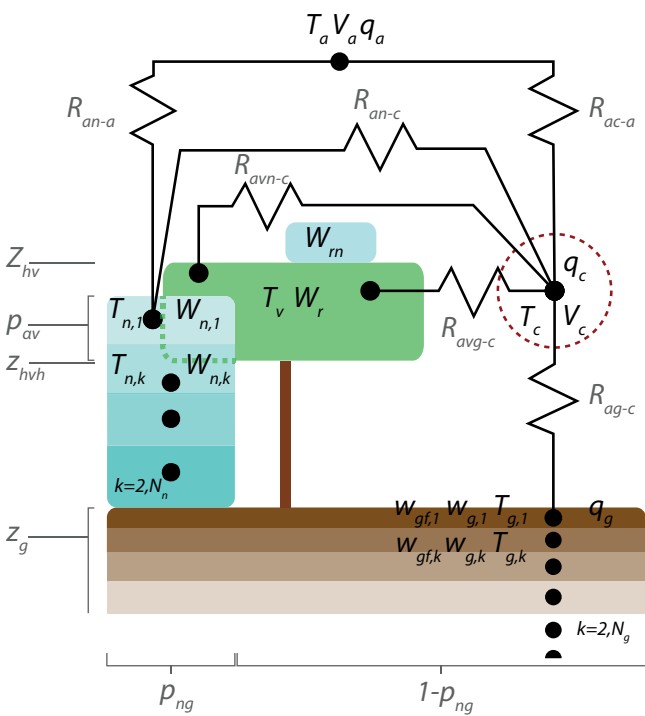

**Figure 1.** A schematic representation of the turbulent aerodynamic resistance, $R_a$, pathways for ISBA-MEB. The prognostic temperature, liquid water, and liquid water equivalent variables are shown. The canopy air diagnostic variables are enclosed by the red-dashed circle. The ground-based snow pack is indicated using turquoise, the vegetation canopy is shaded green, and ground layers are colored brown. Atmospheric variables (lowest atmospheric model or observed reference level) are indicated using the $a$ subscript. The ground snow fraction, $p_{ng}$, and canopy-snow-cover fraction, $p_{n\alpha}$, are indicated.





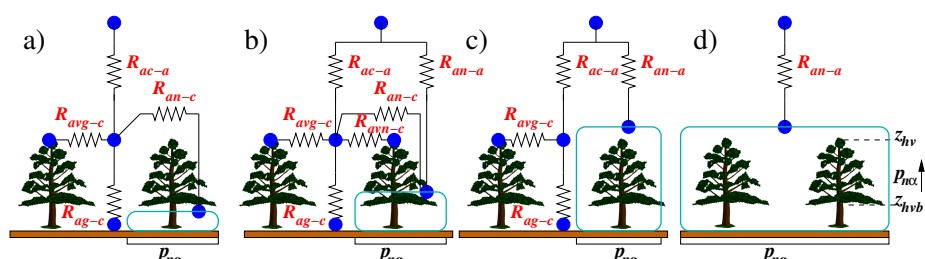

**Figure 2.** A schematic sketch illustrating the role of $p_{n\alpha}$, the fraction of the vegetation canopy which is buried by ground-based snow. In panel a), the snow is well below the canopy base, $z_{hvb}$, resulting in $p_{n\alpha} = 0$ and the snow has no direct energy exchange with the atmosphere. In panel b), the canopy is partly buried by snow ($0 < p_{n\alpha} < 1$) and the snow has energy exchanges with both the canopy air and the atmosphere. In panel c), the canopy is fully buried by snow ($p_{n\alpha} = 1$) and the snow has energy exchange only with the atmosphere while the soil and canopy only exchange with the canopy air space ($p_{ng} < 1$). Finally, in panel d), both $p_{ng} = 1$ and $p_{n\alpha} = 1$, so that the only exchanges are between the snow and the atmosphere.



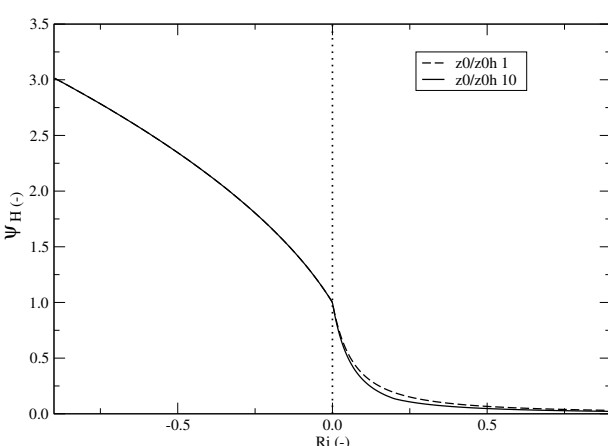

**Figure 3.** Stability correction term is shown using the Sellers formulation for $R_i \leq 0$ while the function for stable conditions adapted from ISBA ($R_i > 0$) for two ratios of $z_{0g}/z_{0gh}$. The ground surface roughness length is defined in Table 2.





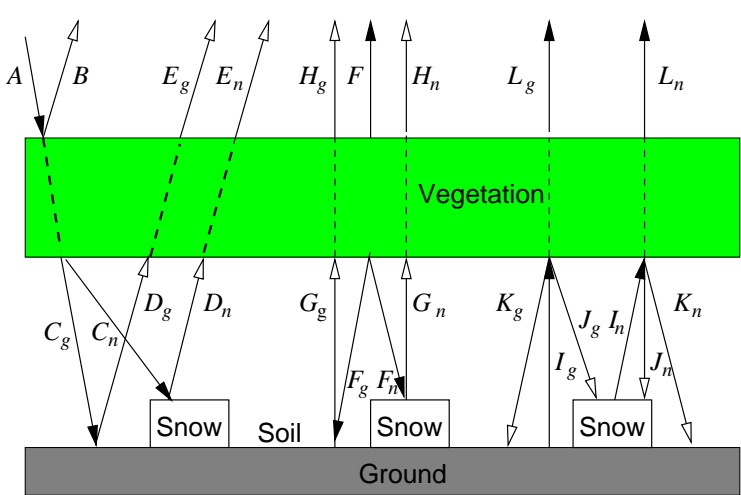

**Figure 4.** Simple schematic for longwave radiation transfer for one reflection and up to three emitting surfaces (in addition to the down-welling atmospheric flux). Hollow arrows indicate fluxes after one reflection.