# Peer review of "The Interactions between Soil-Biosphere-Atmosphere land surface model with a Multi-Energy Balance option (ISBA-MEB) in SURFEXv8 - Part 1: Model description"

_Geoscientific Model Development, 2016_

## Short Comment (SC1) · 30 Nov 2016

Dear authors,

In my role as Executive editor of GMD, I would like to bring to your attention our Editorial version 1.1:

http://www.geosci-model-dev.net/8/3487/2015/gmd-8-3487-2015.html

This highlights some requirements of papers published in GMD, which is also available on the GMD website in the 'Manuscript Types' section:

[Figure]

http://www.geoscientific-model-development.net/submission/manuscript_types.html

In particular, please note that for your paper, the following requirement has not been met in the Discussions paper:

- "The main paper must give the model name and version number (or other unique identifier) in the title."

Please add a version number for ISBA, MEB and SURFEX in the title upon your revised submission to GMD.

Yours,

Astrid Kerkweg

—————————————————

---

## Referee Comment (RC1) · Anonymous Referee #1 · 7 Dec 2016

This paper described an improved version of ISBA for better representation of the land surface processes. I found this paper to be well organized and written and also valuable for the land surface modeling community. Therefore, I propose minor revisions before accepting the paper for publication.

I propose a small change in title "The Interactions between Soil-Biosphere-Atmosphere land surface model with Multi-Energy Balance option (ISBA-MEB) in SURFEX"

P3. L. 63-69: Re-phrase the paragraph describing the different application of ISBA. Possibly break the sentence to multiple sentences each describing one of the ISBA

application. For example "ISBA has been used in operational high resolution short term numerical weather prediction . . ...... It also has been employed in climate research studies . . .."

P3. L75: Remove this phrase "by improved realism with respect to for example" then rephrase to "The force-Restore approach of ISBA has been replaced in recent years by multi layer explicit . . ..."

p3. L90: The last sentence is unfinished!

P4. L93: change "lessening" to reducing or minimizing.

P5. L.130-135: Needs some correction in the text. For example: put parenthesis around "Smith et al. (2011)" and "Zhang et al. (2014)". "over Europe by Wramneby . . .."

P6. L.171: by DIF do you mean ISBA-DF in Boone et al. 2000? what ES stands for? Line 4: is it 12 soil layers? what is the total depth of the soil?

P6. L.188: Define Va

P6. L. 189-191: Re-phrase " equations for the evolutions o the bulk vegetation canopy temperature, Tv, the snow-free ground surface (soil-litter) temperature, Tg, . . ....."

P6. L. 192: change to "equivalent water content of ice" or "ice water equivalent (IWE)"

Fig 1: the colors in the picture is not as indicated in the text.

P8. L 233: should be Eq. 5

P8. L225-227: Define other parameters in Eqs 4-6 (e.g. H, LE, Rn, G, SW) with their units.

P11. L354. Shouldn't be qsatin based on eqs 19, 20?

P16. L490. Are emissivities defined based on the vegetation classes?

P20 L. 637: Correction "soil liquid water content and water content equivalent of frozen water"

**[GMDD](about:blank)**

---

## Referee Comment (RC2) · Anonymous Referee #2 · 9 Dec 2016

This paper describes the introduction of a multi-energy balance version in the ISBA SVAT model. This more complexity-based approach is a major development modification of the model and has the goal to improve the land surface processes, particularly on forest medium. I found this paper well written, well documented and clear. Model history and working context is well introduced. I'm sure that this ISBA option will serve the land surface model community. Therefore, I suggest minor revisions before accepting the paper for publication.

p5. L146: "and" is misplaced

p6. L188: define Va in equation

Fig. 1: this figure is central and commented P.6 from L176 to 201. For a better under-standing of different resistances and temperatures, a table showing symbology indices elements would be welcome such vg: vegetation, c: canopy, g: ground, n: snow surface ...) Moreover, this symbology is repeated in many other terms.

P8. L234:I supposed the reference is Eq. 5 instead of 6

p8. L248: reverse vegetation and snow position.

p10. L321: Snow surface temperature is missing.

p11. L355: replace "over liquid water and ice" by "air and snow".

p13. L393: define "pn" as evaporative efficiency or adapted terminology

p13. L394: define "LAI"

p14. L449: replace "m-6" by "10-6 m" (2 times)

p16. eq. 45: define LAI as Leaf Aera Index

p17. eq. 52: define "lw" (denominator)

p18. eq. 58: is LAIf a particular LAI?

p22. L698: boeotian question: I supposed the maximum snow load per unit branch area is different according to species. Is value of 6.3 kg m-2 proposed could be consider as a median estimator? or a default value?
* * *

---

## Author Comment (AC1) · 9 Jan 2017

COMMENT: "The main paper must give the model name and version number (or other unique identifier) in the title."

RESPONSE: this information is included in Section 4 (Code Availability)

COMMENT: Please add a version number for ISBA, MEB and SURFEX in the title upon your revised submission to GMD.

RESPONSE: We have added this to the title (SURFEXv8). ISBA and MEB versions

evolve within each SURFEX release, so changes in ISBA and MEB are reflected by the SURFEX version number.

---

## Author Response (AR1)

[revised manuscript text omitted]

**Authors: A. Boone, P. Samuelsson, S. Gollvik, A. Napoly, L. Jarlan, E. Brun and B. Decharme**

**Corresponding Author response to Anonymous Referee #1**

Comment:
I propose a small change in title "The Interactions between Soil-Biosphere-Atmosphere
land surface model with Multi-Energy Balance option (ISBA-MEB) in SURFEX"

Response:
In accordance with this reviewers suggestion, we have modified the title to:

"The Interactions between Soil-Biosphere-Atmosphere land surface model with a Multi-Energy
Balance (ISBA-MEB) option in SURFEXv8 - Part 1: Model description"

Please note that we retained the text - "Part 1: Model description" – since it is the first of a 2 part
series. But, we will leave the decision on whether or retain or not this last phrase up to the editor.

Comment:
P3. L. 63-69: Re-phrase the paragraph describing the different application of ISBA. Possibly break
the sentence to multiple sentences each describing one of the ISBA application. For example "ISBA
has been used in operational high resolution short term numerical weather prediction . . ..... It also
has been employed in climate research studies . . .."

Response:
Done: this paragraph has been edited to split a single long sentence into multiple sentences. (new
L.62-70)

Comment:
P3. L75: Remove this phrase "by improved realism with respect to for example" then rephrase to
"The force-Restore approach of ISBA has been replaced in recent years by multi layer explicit . . ..."

Response:
Done in new L.76-77

Comment:
3. L90: The last sentence is unfinished!

Response:
It has been rewritten (and completed) as:

in order to distinguish the soil, snow and vegetation surface temperatures since they can have very
different amplitudes and phases in terms of the diurnal cycle. Accounting for this distinction
facilitates (at least conceptually) incorporating remote-sensing data, such as satellite-based thermal
infrared temperatures (Anderson_ea_97), into such models.
(new L.89-92)

Comment:
P4. L93: change "lessening" to reducing or minimizing.

Response:
Done: new L.94

Comment:
P5. L.130-135: Needs some correction in the text. For example: put parenthesis around "Smith et al. (2011)" and "Zhang et al. (2014)". "over Europe by Wramneby. ..."

Response:
Done: new L.132-135

Comment:
P6. L.171: by DIF do you mean ISBA-DF in Boone et al. 2000? what ES stands for?
Line 4: is it 12 soil layers? what is the total depth of the soil?

Response:
We have now defined the acronyms DF (DiFfusion equation) and ES (Explicit Snow process) in the text, and the default number of layers (14 and 12, respectively, for each). See new L.172-174

Comment:
P6. L.188: Define Va

Response:
Va=the wind speed at the atmospheric forcing level (represented by the subscript "a"): this is now defined in the text. New L.196-197

Comment:
P6. L. 189-191: Re-phrase " equations for the evolutions o the bulk vegetation canopy temperature, Tv, the snow-free ground surface (soil-litter) temperature, Tg, . . ....."

Response:
Done: reworded as:
"The surface energy budgets are formulated in terms of prognostic equations governing the evolutions of the bulk vegetation canopy….". new L.198-200

Comment:
P6. L. 192: change to "equivalent water content of ice" or "ice water equivalent (IWE)"

Response:
Done: we have used the former choice of wording. New L.201

Comment:
Fig 1: the colors in the picture is not as indicated in the text.

Response:
We have corrected this: we verified the caption (it is OK) and have edited the text in the body of the paper (near the beginning of Section 2 where the Fig. 1 is referred to) to be consistent with both the caption and  Fig.1.
new L.184-186

Comment:
P8. L 233: should be Eq. 5

Response:
In fact, this should be Eq.6. (although indeed it is also applicable to Eq.5). But we can see this has lead to some confusion: Although $p_{ng}$ cancels in Eq.6, it has been used here simply because by multiplying by $p_{ng}$, energy conservation can be obtained by summing Eq.s4-6. But indeed we have realized that this is a bit awkward, thus we have dropped $p_{ng}$ in Eq.6 (essentially it cancels out since it appears on both the RHS and LHS of Eq.6, so in fact this represents no change to the math). The same is true for the discretized forms of the snow heat and mass prognostic equations...i.e. Eq.s G2 and G11 (so for consistency, we have also canceled out $p_{ng}$ from both sides of those equations). We now emphasize later in the paper that when combining Eq.s 4-6 (to solve them simultaneously and for mass/energy conservation of the entire patch or grid cell), we must multiply Eq.6 by $p_{ng}$ (specifically, we emphasize this now in the Appendices, notably G, just after Eq.G3 and G14, and I, after Eq. I3). Note that dropping $p_{ng}$ from Eq.s6, G2 and G11 does not cause any of the other derivations/equations to change. Finally, we edited the text to reflect the reviewers main comment : that indeed the text can apply to Eq.5.
See new L.243-249

Comment:
P8. L225-227: Define other parameters in Eqs 4-6 (e.g. H, LE, Rn, G, SW) with their units.

Response:
The terms were defined in successive parts of the text when the corresponding mathematical definitions are given: but indeed defining them upon first appearance is a good suggestion (improving readability and more standard) so we have done as the reviewer requests : we have moved the definitions and adjusted the text accordingly.
New L.240-252.

Comment:
P 11. L354. Shouldn't be qsatin based on eqs 19, 20?

Response:
Yes: but in fact a naming convention is defined in the text just after this equation so we didn't define it explicitly: but we realize based on this reviewers comment that it is not explained well enough so we have added the text giving the explicit definition:
The same convention holds for saturation over ice, so that $q_{satin}$ represents the value over the snowpack.
New L.370-374

Comment:
P16. L490. Are emissivities defined based on the vegetation classes?

Response:
Yes: emissivities depend mainly upon vegetation class. But some dependence on climate (location) exists, but indeed they depend primarily on vegetation class: the text has been modified to add this information.
New L. 508-509

Comment:
P20 L. 637: Correction "soil liquid water content and water content equivalent of frozen water"

Response:
Correction done. New L.657-658

**REF: Geosci. Model Dev. Discuss., doi:10.5194/gmd-2016-269, 2016.**
**Title: "The Interactions between Soil-Biosphere-Atmosphere (ISBA) land surface model Multi Energy Balance (MEB) option in SURFEX – Part 1: Model description"**
**Authors: A. Boone, P. Samuelsson, S. Gollvik, A. Napoly, L. Jarlan, E. Brun and B. Decharme**

**Corresponding Author response to Anonymous Referee #2**

Comment:
p5. L146: "and" is misplaced

Response:
Done: reworded as a new phrase using a break as:
...and shaded leaves.
It was primarily developed to improve the modeling of photosynthesis within ISBA…
New L.146-147.

Comment:
p6. L188: define Va in equation

Response:
Va=the wind speed at the atmospheric forcing level (represented by the subscript "a"): this is now defined in the text. New L.196-197

Comment:
Fig. 1: this figure is central and commented P.6 from L176 to 201. For a better understanding of different resistances and temperatures, a table showing symbology indices elements would be welcome such vg: vegetation, c: canopy, g: ground, n: snow surface ...) Moreover, this symbology is repeated in many other terms.

Response:
We have done this and the new table is now labeled as Table 2. (thus the Table previously labeled as 2 is now 3). All of the symbols for distinguishing between prognostic and diagnostic variables, and the aerodynamics resistances are listed and described. This table is referred to right after Fig.1 is first mentioned in the text.
New L.184 (and new Table2)

Comment:
p8. L234:I supposed the reference is Eq. 5 instead of 6

Response:
In fact, this should be Eq.6. (although indeed it is also applicable to Eq.5). But we can see this has lead to some confusion: Although p_ng cancels in Eq.6, it has been used here simply because by multiplying by p_ng, energy conservation can be obtained by summing Eq.s4-6. But indeed we have realized that this is a bit awkward, thus we have dropped p_ng in Eq.6 (essentially it cancels out since it appears on both the RHS and LHS of Eq.6, so in fact this represents no change to the math). The same is true for the discretized forms of the snow heat and mass prognostic equations...i.e. Eq.s G2 and G11 (so for consistency, we have also canceled out p_ng from both sides of those equations). We now emphasize later in the paper that when combining Eq.s 4-6 (to solve them simultaneously and for mass/energy conservation of the entire patch or grid cell), we must multiply Eq.6 by p_ng  (specifically, we emphasize this now in the Appendices, notably G, just after Eq.G3 and G14, and I, after Eq. I3). Note that dropping p_ng from Eq.s6, G2 and G11 does not cause any of the other derivations/equations to change. Finally, we edited the text to reflect

the reviewers main comment : that indeed the text can apply to Eq.5.
See new L.243-249

Comment:
p10. L321: Snow surface temperature is missing.

Response:
We have clarified this by adding the temperatures for the bulk vegetation, ground surface and snow surface to this line.
New L.341-342

Comment:
p11. L355: replace "over liquid water and ice" by "air and snow".

Response:
$q\_sat$ represents the saturation specific humidity over liquid water (generally speaking, for v, g)...$q\_sati$ represents that over ice (for snow). But this section was apparently not very clear (the other reviewer also requested some clarification), thus we have rewritten the text between Eq.s21-22 to more clearly define the different $q\_sat$ definitions.
New L.370-374

Comment:
p13. L393: define "pn" as evaporative efficiency or adapted terminology

Response:
The $p\_nv$ is now defined in the text:
where $p_{nv}$ is an evaporative efficiency factor which is used to partition the canopy interception storage mass flux between evaporation of liquid water and sublimation
New L.411-412

Comment:
p13. L394: define "LAI"

Response:
We remove LAI from the dicussion here since it doesn't appear in the expressions. We defined it now just after Eq.45 (in response to this reviewer's comment below: comment after the next)
New L. 519-520

Comment:
p14. L449: replace "m-6" by "10-6 m" (2 times)

Response:
Done. New L.466-467

Comment:
p16. eq. 45: define LAI as Leaf Aera Index

Response:
we have added the definition of LAI here as requested
New L. 519-520

Comment:

p17. eq. 52: define "lw" (denominator)

Response:
Done: lw=leaf width (m)
New L.555

Comment:
p18. eq. 58: is LAIf a particular LAI?

Response:
No, in fact this is a typo, it has now been corrected as LAI
New L.581 (Eq.58)

Comment:
p22. L698: boeotian question: I supposed the maximum snow load per unit branch area is different according to species. Is value of 6.3 kg m-2 proposed could be consider as a median estimator? or a default value?

Response:
In fact, 6.3 kg m-2 is an average value for both pine (6.6) and spruce (5.9) based on measurements from Schmidt and Gluns (1991). Thus, over these two different (and fairly representative forest types), the average value for this parameter only varies by about 10%. Thus, we currently use 6.3 as the default value for all species. The above has now been included in the text cited above.
New L. 718-722

---

## Author Response (AR2)

REF: Geosci. Model Dev. Discuss., doi:10.5194/gmd-2016-269, 2016. Title: "The Interactions between Soil-Biosphere-Atmosphere (ISBA) land surface model Multi Energy Balance option (MEB) in SURFEX – Part 1: Model description" Authors: A. Boone, P. Samuelsson, S. Gollvik, A. Napoly, L. Jarlan, E. Brun and B. Decharme

Corresponding Author response to Editor: From the editor:
Thank you for this careful revision of your manuscript answering basically all the referees' comments. I however have the minor following remarks to make:

COMMENT:
-Paper title: I think it would read better to put "option" before "(ISBA-MEB)".

RESPONSE:
The title has now been modified accordingly.

COMMENT:
-L12-13: wouldn't it be better to put this sentence at the past tense: "It became clear ... had been reached and there was a need ..." ?

RESPONSE:
This sentence has now been edited/corrected to be in the past tense.

COMMENT:
-L23: please put a comma after "part two": part two, which …

RESPONSE:
Corrected/done

COMMENT:
-L76-77: you write that you changed L76-77 to answer the referee' comment but I do not see any changes and the sentence still reads "... has been replaced in recent years by improved realism with respect to ...".

RESPONSE:
Indeed, this was an oversight (Referee 1 comment). We have now modified the phrase as suggested by that reviewer:

«In the SURFEX context, the Force-Restore approach has been replaced in recent years by multi-layer explicit physically-based options for...»

COMMENT:
-L170-175: Again, I think that "... ISBA-MEB option (referred to hereafter simply as MEB) ..." and "... multi-layer soil scheme (ISBA-DF: explicit DiFfusion equation for heat and Richard's equation for soil water flow) ..." and "snow parameterisation (ISBA-ES: multi-layer Explicit Snow processes with 12 layers by default) ..." would read better.

RESPONSE:
We are not quite sure what the requested modification is: but we have slightly reworded the phrases: we hope that this is along the lines of what the editor meant. The new text reads:

The ISBA-MEB (referred to hereafter simply as MEB) option can be activated for any number of the forest patches. By default, MEB is coupled to the multi-layer soil (ISBA-DF: explicit DiFfusion equation for heat and Richard's equation for soil water flow, Boone et al. 2000; Decharme et al., 2011) and snow (ISBA-ES: multi-layer Explicit Snow processes with 12 layers by default, Boone and Etchevers 2001; Decharme et al., 2016) schemes.

COMMENT:
-L657-658: I do not see the correction and the sentence "The soil liquid water and equivalent frozen water equivalent volumetric water content are defined ..." still sounds awkward to me.

RESPONSE:
We have replaced this phrase with the suggestion by Reviewer 1 to now read:

«The soil liquid water content and water content equivalent of frozen water are defined as w_g and w_gf, respectively (m³ m-³). «